# Adversarial Attacks on Data Attribution

**Xinhe Wang**
University of Michigan
sevenmar@umich.edu

**Pingbang Hu**
University of Illinois Urbana-Champaign
pbb@illinois.edu

**Junwei Deng**
University of Illinois Urbana-Champaign
junweid2@illinois.edu

**Jiaqi W. Ma**
University of Illinois Urbana-Champaign
jiaqima@illinois.edu

## Abstract

Data attribution aims to quantify the contribution of individual training data points to the outputs of an AI model, which has been used to measure the value of training data and compensate data providers. Given the impact on financial decisions and compensation mechanisms, a critical question arises concerning the adversarial robustness of data attribution methods. However, there has been little to no systematic research addressing this issue. In this work, we aim to bridge this gap by detailing a threat model with clear assumptions about the adversary's goal and capabilities and proposing principled adversarial attack methods on data attribution. We present two methods, *Shadow Attack* and *Outlier Attack*, which generate manipulated datasets to inflate the compensation adversarially. The Shadow Attack leverages knowledge about the data distribution in the AI applications, and derives adversarial perturbations through "shadow training", a technique commonly used in membership inference attacks. In contrast, the Outlier Attack does not assume any knowledge about the data distribution and relies solely on black-box queries to the target model's predictions. It exploits an inductive bias present in many data attribution methods—outlier data points are more likely to be influential—and employs adversarial examples to generate manipulated datasets. Empirically, in image classification and text generation tasks, the Shadow Attack can inflate the data-attribution-based compensation by at least $200\%$, while the Outlier Attack achieves compensation inflation ranging from $185\%$ to as much as $643\%$. Our implementation is ready at https://github.com/TRAIS-Lab/adversarial-attack-data-attribution.

## 1 Introduction

*Data attribution* aims to quantify the contribution of individual training data points to the outputs of an Artificial Intelligence (AI) model (Koh & Liang, 2017). A key application of data attribution is to measure the value of training data in AI systems, enabling appropriate compensation for data providers (Ghorbani & Zou, 2019; Jia et al., 2019). With the rapid advancement of generative AI, these methods have gained increased relevance, particularly in addressing copyright concerns. Recent studies have explored economic frameworks using data attribution for copyright compensation, showing promising preliminary results (Deng & Ma, 2023; Wang et al., 2024).

Given the significant potential of data attribution methods for data valuation and compensation, an important question arises regarding their adversarial robustness. As these methods influence financial decisions and compensation mechanisms, they may attract malicious actors seeking to manipulate the system for personal gain. This underscores the need to investigate whether data attribution methods can be manipulated and exploited, as their vulnerabilities could lead to unfair compensation, undermining the trustworthiness of the solutions built on top of these methods.

However, adversarial attacks on data attribution methods have received little to no exploration in prior literature. Along with the proposal of a data-attribution-based economic solution for copyright compensation, Deng & Ma (2023) briefly experimented with a few heuristic approaches (e.g., dupli-

cating data samples) for attacking data attribution methods. A systematic study that clearly defines the threat model and develops principled adversarial attack methods has yet to be conducted.

This work presents the first comprehensive study to fill this gap. We first outline the threat model by detailing the data compensation workflow and specifying the assumptions we made. One key assumption is that the data contribution is periodic, and there is certain persistence across consecutive iterations of data contributions, which is the source of knowledge that the adversary could exploit. We also assume that the adversary may either have access to the distribution of the data used by the target model of the AI system or can get black-box queries of the target model's predictions, both are commonly seen in the AI security literature (Shokri et al., 2017; Chen et al., 2017).

Subsequently, we propose two adversarial attack strategies, *Shadow Attack* and *Outlier Attack*, respectively relying on different assumptions about the adversary's capabilities. The Shadow Attack relies on the access to data distribution and employs the "shadow training" technique commonly used in membership inference attacks (Shokri et al., 2017) to train "shadow models" that imitate the target model. The adversary can then directly perturb their dataset to achieve a higher compensation on these shadow models. The Outlier Attack, instead, does not assume knowledge about the data distribution but only relies on black-box queries of the target model's predictions. The key idea behind this method lies in an inductive bias of many data attribution methods—outlier data points are more likely to be more influential. The proposed Outlier Attack utilizes adversarial examples (Goodfellow et al., 2015; Chen et al., 2017) to generate realistic outliers in a black-box fashion.

We conduct extensive experiments, including both image classification and text generation settings, to demonstrate the effectiveness of the proposed attack methods. Our results show that by only adding imperceptible perturbations to real-world data features, the Shadow Attack can inflate the adversary's compensation to at least $200\%$ and up to $456\%$, while the Outlier Attack can inflate the adversary's compensation to at least $185\%$ and up to $643\%$.

Overall, our study reveals a critical practical challenge—adversarial vulnerability—in deploying data attribution methods for data valuation and compensation. Moreover, the design of the proposed attack methods, especially the Outlier Attack that exploits a common inductive bias of data attribution methods, offers deeper insights into these vulnerabilities. These findings provide valuable directions for future research to enhance the robustness of data attribution methods.

## 2  RELATED WORK

**Data Attribution for Data Valuation and Compensation.**  Data attribution methods have been widely used for quantifying the value of training data in AI applications and compensating data providers (Ghorbani & Zou, 2019; Jia et al., 2019; Yoon et al., 2020; Kwon & Zou, 2022; Feldman & Zhang, 2020; Xu et al., 2021; Lin et al., 2022; Kwon & Zou, 2023; Just et al., 2023; Wang & Jia, 2023; Deng & Ma, 2023; Wang et al., 2024). With the rapid advancement of generative AI, these methods have gained increasing relevance due to the growing concerns around copyright. Recent studies have proposed economic solutions using data attribution for copyright compensation, yielding promising preliminary results (Deng & Ma, 2023; Wang et al., 2024). Given the significant potential of data attribution methods for data valuation and compensation, a critical question arises regarding their adversarial robustness. Aside from one earlier exploration using heuristic approaches to attack data attribution methods (Deng & Ma, 2023), no systematic study has addressed this issue. This work presents the first comprehensive study that outlines a detailed threat model and proposes principled and effective approaches for adversarial attacks on data attribution methods.

**Membership Inference Attack.**  Membership inference attacks aim to infer whether a specific data point was used during the training of a machine learning model, typically without knowing the actual training dataset or having white-box access to the model (Shokri et al., 2017). The general strategy involves leveraging information such as model architecture, training data distribution, or black-box model predictions (Shokri et al., 2017; Song & Mittal, 2021). We refer the readers to Hu et al. (2022) for a detailed survey on this topic. One of the proposed attack methods, the Shadow Attack, is inspired by the "shadow training" technique (Shokri et al., 2017) commonly used in membership inference attacks, where the adversary draws "shadow samples" following the same distribution as the actual training dataset used by the target model to be attacked, and trains "shadow models" based on the shadow samples to imitate the target model.

**Adversarial Example.** Adversarial example (Goodfellow et al., 2015) is a well-known phenomenon where small, often imperceptible perturbations to input features can significantly alter the predictions of machine learning models, particularly deep neural networks. These perturbations can be generated through black-box queries to the model predictions (Chen et al., 2017; Ilyas et al., 2018; Guo et al., 2019). For a comprehensive review of adversarial examples, see Yuan et al. (2019); Chakraborty et al. (2021). In the proposed Outlier Attack, we employ black-box adversarial attack methods for generating adversarial examples to generate realistic outliers relative to the training dataset of the target model, without needing access to the training dataset or the model details.

## 3 THE THREAT MODEL

### 3.1 THE DATA COMPENSATION SCENARIO

We consider a scenario where there is an `AI Developer`, and a (potentially large) set of `Data Providers`. The `Data Providers` supply training data for the `AI Developer` to develop an AI model. In return, the `Data Providers` are compensated based on their data's contribution to the model, as measured by a specific data attribution method.

**Periodic Data Contribution.** We assume that the `Data Providers` contribute data to the `AI Developer` *periodically*, a common practice in many AI applications. For example, large language models need periodic updates to stay aligned with the latest factual knowledge about the world (Zhang et al., 2023); recommender systems must adapt to evolving user preferences (Zhang et al., 2020); quantitative trading firms rely on up-to-date information to power their predictive models[1]; and generative models for music or art benefit from fresh, innovative works by artists to diversify their creative outputs (Smith et al., 2024). However, such periodic data contribution introduces risks: a malicious `Data Provider` (referred to as an `Adversary` thereafter) could exploit information from previous iterations to adversarially manipulate their future data contribution, potentially inflating their compensation unfairly.

To formalize this scenario, without loss of generality, we consider two consecutive iterations of data contribution, denoted as time steps $t = 0$ and $t = 1$. At $t = 0$, there is no `Adversary` and the training dataset consists solely of contributions from benign `Data Providers`. This dataset is represented as $Z_0 \in \mathcal{Z}$, where $\mathcal{Z}$ is the set of all possible training datasets. At $t = 1$, the training dataset is $Z_1 = Z_1^b \cup Z_1^a$, where $Z_1^b \in \mathcal{Z}$ represents the training data provided by benign `Data Providers`, while $Z_1^a \in \mathcal{Z}$ is the set provided by the `Adversary`. We also assume that $Z_1^b \cap Z_1^a = \varnothing$, meaning there is no overlap between the two datasets. Finally, for a dataset $Z \in \mathcal{Z}$, each data point $z \in Z$ is represented as a pair $z = (x, y)$, where $x \in \mathcal{X}$ is the input feature and $y \in \mathcal{Y}$ is the prediction target, with $\mathcal{X}$ and $\mathcal{Y}$ referring to the feature space and target space, respectively.

**AI Training and Data Attribution.** Let $\mathcal{M}$ represent the set of AI models, and let $\mathcal{T} \colon \mathcal{Z} \to \mathcal{M}$ denote a *training algorithm*, mapping a training dataset $Z \in \mathcal{Z}$ to a model $\mathcal{T}(Z) \in \mathcal{M}$.

In data attribution, we aim to understand how individual data points from a training dataset $Z \in \mathcal{Z}$ contribute to the model output on a target (validation) data point from a validation dataset $V$. Given $Z$ and $V$, a *data attribution method* derives a *contribution function* $\tau \colon Z \times V \to \mathbb{R}$ that assigns a real value $\tau(z, v)$ to each training data point $z \in Z$ for a given validation data point $v \in V$. Denote the set of all such function $\tau$'s as $\mathcal{C}$, and the set of all possible validation sets as $\mathcal{V}$. Therefore, a data attribution method can be formalized as a function $\mathcal{A} \colon \mathcal{Z} \times \mathcal{M} \times \mathcal{V} \to \mathcal{C}$. In most cases, we will consider $\mathcal{A}(Z, \mathcal{T}(Z), V)$, hence $Z$ and $V$ alone suffice to specify the resulting $\tau$.

**Compensation Mechanism.** In practice, the contribution function $\tau$ derived from data attribution methods may not reliably measure the contributions of all training data points due to the inherent randomness in AI model training and the need for efficiency (Wang & Jia, 2023; Nguyen et al., 2024). Specifically, measurements of data points with smaller contributions are often less reliable than those of the most influential contributors. Following Deng & Ma (2023), we consider a compensation mechanism where only the top-$k$ influential training data points for each validation data point $v \in V$ receive a fixed amount of compensation.

---

[1]See, for example, the Bloomberg market data feed: `https://www.bloomberg.com/professional/products/data/enterprise-catalog/market/`.

## 3.2 THE ADVERSARY

We now discuss the `Adversary`'s objective and capability under the data compensation scenario.

**The Objective of the Adversary.** Let $V_1$ be the validation dataset used at step $t = 1$. The objective of the `Adversary` is to construct a dataset $Z_1^a$ that maximizes the *compensation share* received by the `Adversary`, which is defined as

$$c(Z_1^a) = \frac{1}{k|V_1|} \sum_{z \in Z_1^a} \sum_{v \in V_1} \mathbf{1}[\tau_1(z, v) \in \text{Top}_k (\{\tau_1(z', v) \mid z' \in Z_1\})], \tag{1}$$

where $\tau_1 = \mathcal{A}(Z_1, \mathcal{T}(Z_1), V_1)$ is the contribution function at $t = 1$; $\text{Top}_k(\cdot)$ extracts the top-$k$ elements from a finite set of real numbers; and $\mathbf{1}[\cdot]$ is the indicator function.

**The Capabilities of the Adversary.** We first outline the limitations imposed on the `Adversary`. In realistic scenarios, the `Adversary` does **not** have access to any of the following:

- the exact training datasets ($Z_0$ and $Z_1^b$) and the exact validation datasets ($V_0$ and $V_1$);
- white-box access to the trained models $\mathcal{T}(Z_0)$ and $\mathcal{T}(Z_1)$;
- the contribution functions $\mathcal{A}(Z_0, \mathcal{T}(Z_0), V_0)$ and $\mathcal{A}(Z_1, \mathcal{T}(Z_1), V_1)$.

Then we make the following assumptions to characterize the capabilities of the `Adversary`.

**Assumption 1** (Persistence). *Assume that $Z_0 \subseteq Z_1^b$ and $|Z_0|/|Z_1^b|$ is close to 1. Additionally, assume that $V_0$ and $V_1$ are independently sampled from the same distribution.*

**Assumption 2** (Access to data distribution and training algorithm). *Assume that the `Adversary` has access to the distributions of $Z_0$ and $V_0$. Additionally, assume that the `Adversary` has knowledge about the training algorithm $\mathcal{T}$ (but not the model $\mathcal{T}(Z_0)$).*

**Assumption 3** (Black-box access to model). *Assume that the `Adversary` has access to black-box query to the model $\mathcal{T}(Z_0)$ to query the model predictions on any input feature $x \in \mathcal{X}$.*

Intuitively, Assumption 1 assumes persistence between time steps $t = 0, 1$, so that information gained from $t = 0$ can inform the attack at $t = 1$. In Assumption 2 and Assumption 3, it is worth noting that the `Adversary`'s access to information is limited to what is available at $t = 0$ only.

In practice, Assumption 1 is realistic in many real-world applications. For example, in applications that have periodic model updates, such as large language models or recommender systems, a substantial portion of the training data often remains consistent across consecutive iterations, with only incremental changes. Moreover, Assumption 2 is a common assumption in the membership inference attack literature (Shokri et al., 2017), while Assumption 3 reflects a common setup for generating adversarial examples against neural network models (Chakraborty et al., 2021).

The two proposed attack methods in Section 4 and Section 5 respectively rely on Assumption 2 and Assumption 3, but do not depend on both simultaneously. As a result, the first method is a gray-box attack method while the second method is a black-box attack method.

**The Action Space of the Adversary.** We assume that the `Adversary` is restricted to making only small perturbations to an existing set of real data points to construct the adversarial dataset $Z_1^a$. This implies that the `Adversary` cannot introduce entirely synthetic or arbitrary data, but rather, can modify real data points subtly. This reflects a realistic adversarial scenario as overly large or unnatural alterations would be easily detectable.

## 4 SHADOW ATTACK

In this section, we introduce the proposed *Shadow Attack*, which leverages Assumption 1 and Assumption 2, and exploits the knowledge about the data distribution at $t = 0$ to perform attacks.

At a high level, the `Adversary` first performs a *shadow training* process, where models are trained on data drawn from a distribution similar to that of the training dataset $Z_0$, allowing the `Adversary` to approximate the target model $\mathcal{T}(Z_0)$. The `Adversary` then applies adversarial perturbations to the data points they plan to contribute to the `AI Developer` at $t = 1$. The adversarial perturbations are derived by maximizing the data attribution values on the models obtained through shadow training. See Figure 1 for an illustration.

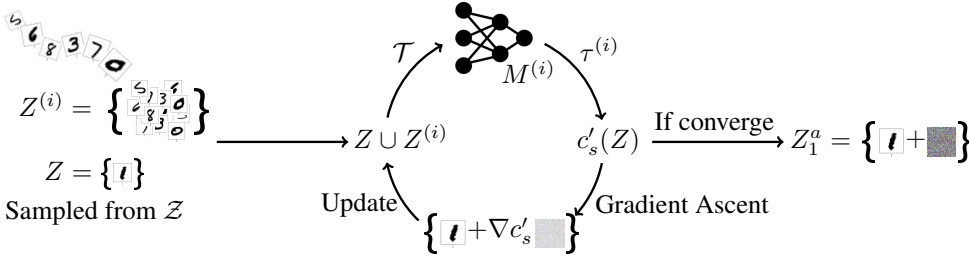

Figure 1: An illustration of the Shadow Attack method. Shadow training datasets $Z^{(i)}$'s are sampled to estimate the compensation share of a set of data points $Z$ if it were contributed to the `AI Developer`, which can be leveraged to perturb data points in $Z$ to get a higher compensation share.

## 4.1 Shadow training

Given a dataset $Z \in \mathcal{Z}$ (that the `Adversary` may eventually contribute to the `AI Developer` as $Z_1^a$), the goal of the shadow training process is to estimate the data attribution values of the elements in $Z$ as if this dataset were contributed to the `AI Developer` at $t = 1$.

We first sample $m$ shadow training datasets, denoted as $Z^{(1)}, Z^{(2)}, \ldots, Z^{(m)}$. These datasets are independent of the actual training dataset used by the `AI Developer` but follow the same distribution. For each shadow training dataset $Z^{(i)}, i = 1, \ldots, m$, we train a corresponding shadow model $M^{(i)} \in \mathcal{M}$. The shadow model $M^{(i)}$ is trained on $Z^{(i)} \cup Z$ using the same training algorithm $\mathcal{T}$ as by the `AI Developer`.[2] i.e., $M^{(i)} = \mathcal{T}(Z^{(i)} \cup Z)$.

In order to estimate the data attribution values, we further sample a shadow validation dataset $V^{(0)}$ following the same distribution as $V_0$. We can estimate a shadow contribution function using each shadow training dataset $Z^{(i)} \cup Z$, the corresponding shadow model $M^{(i)}$, and the shadow validation dataset $V^{(0)}$, resulting in $\tau^{(i)} = \mathcal{A}(Z^{(i)} \cup Z, M^{(i)}, V^{(0)})$. Similar to Eq. (1), we consider the following shadow compensation share for the dataset of interest $Z$:

$$c_s(Z) = \frac{1}{mk|V^{(0)}|} \sum_{i=1}^{m} \sum_{z \in Z} \sum_{v \in V^{(0)}} \mathbf{1} \left[ \tau^{(i)}(z, v) \in \text{Top}_k(\{\tau^{(i)}(z', v) \mid z' \in Z^{(i)} \cup Z\}) \right]. \quad (2)$$

## 4.2 Adversarial manipulation by maximizing shadow compensation rate

Given the shadow compensation rate, the natural idea is to apply adversarial perturbations to $Z$ by solving $Z_1^a = \arg\max_{Z' \in \mathcal{N}(Z)} c_s(Z')$, where $\mathcal{N}(Z)$ specifies the space of datasets with undetectable perturbations. In practice, however, solving this optimization problem has two technical challenges. Firstly, the objective $c_s(\cdot)$ is discrete and can be difficult to optimize. Secondly, when doing iterative-style optimization, such as gradient ascent, $\tau^{(i)}$'s need to be updated and re-evaluated at every step as it depends on $Z$ and hence $M^{(i)} = T(Z^{(i)} \cup Z)$. This leads to two problems: On the one hand, updating $M^{(i)}$'s requires retraining, which is computationally heavy; on the other hand, many data attribution methods are computationally expensive, even with all the data and (retrained) models available. Hence, maximizing $c_s(\cdot)$ requires repeatedly retraining and running data attributions on perturbed datasets, which may be infeasible for many cases.

To address the first challenge, we replace $c_s(\cdot)$ with the following surrogate objective

$$c_s'(Z) = \frac{1}{mk|V^{(0)}|} \sum_{i=1}^{m} \sum_{z \in Z} \sum_{v \in V^{(0)}} \tau^{(i)}(z, v), \quad (3)$$

which aims to directly maximize the contribution values of the data points in $Z$ on the shadow validation set $V^{(0)}$. To address the second challenge, we first approximate the retrained models by the initial models trained with the original $Z$ (before any gradient ascent steps) for computational

---

[2]In our experiment in Section 6.2, we demonstrate that the Shadow Attack remains effective when the shadow models have a slightly different architecture compared to the target model.

efficiency, and we further adopt *Grad-Dot* (Charpiat et al., 2019), one of the most efficient data attribution methods when evaluating the contribution function.[3] Using this method, the contribution value $\tau^{(i)}(z, v)$ for any training data point $z$ and validation data point $v$ equals the dot product between the gradients of the loss function on $z$ and that on $v$, evaluated with model $M^{(i)}$.

With such simplification, the adversarial perturbation can be derived by maximizing Eq. (3), which can be solved efficiently through gradient ascent. Note that since Eq. (3) is a constrained optimization problem, when $\mathcal{N}(Z)$ is bounded, gradient ascent is guaranteed to converge to some local optimums. In practice, we perform a fixed number of iterations and carefully control the overall perturbation budgets when solving Eq. (3). The computation cost for each iteration is approximately the same as one forward and one backward pass on each of the $m$ shadow models.

## 5    OUTLIER ATTACK

For large-scale AI applications such as generative AI services, Assumption 2 might be overly strong as it could be difficult for the `Adversary` to guess the distribution of the full training data. However, in this case, Assumption 3 often holds. In this section, we further propose *Outlier Attack*, which only relies on Assumption 1 and Assumption 3. See Figure 2 for an illustration.

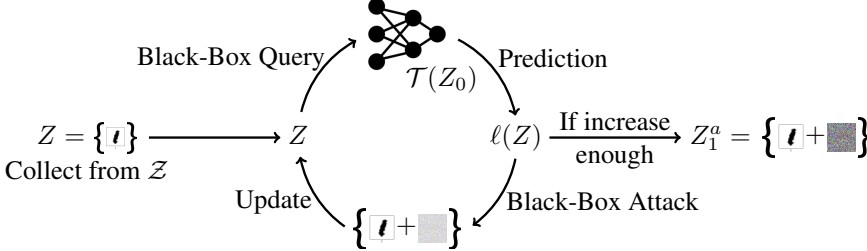

Figure 2: An illustration of the Outlier Attack method. Here, $\ell(Z)$ denotes the loss used by the model $\mathcal{T}(Z_0)$ when evaluated on the dataset $Z$. The data points in $Z$ are perturbed by maximizing the loss $\ell(Z)$ through black-box attack methods designed to generate adversarial examples.

### 5.1    THE OUTLIER INDUCTIVE BIAS OF DATA ATTRIBUTION

The core idea behind Outlier Attack leverages an inherent inductive bias present in many data attribution methods: outlier data points in the training dataset tend to be more influential. Indeed, one of the very first applications of the Influence Function developed in the statistic literature was to detect outliers in training data (Cook, 1977). Consequently, if the `Adversary` contributes a set of outlier data points, they are more likely to get higher compensation, especially given that the compensation mechanism focuses on the top influential data points.

However, translating this intuition into a practical adversarial manipulation strategy poses two significant challenges. Firstly, the contributed data points must closely resemble real-world data; otherwise, they could be easily flagged by data quality filters employed by the `AI Developer`. Secondly, we aim to develop an attack method that does not rely on direct knowledge about the training dataset or its underlying distribution. This makes it challenging to determine whether certain data points truly qualify as outliers relative to the training dataset gathered by the `AI Developer`.

### 5.2    GENERATING REALISTIC OUTLIERS WITH ADVERSARIAL ATTACKS

To tackle the first challenge, we propose starting with a set of real-world data and transforming them into outliers through small adversarial perturbations. With careful design, this strategy also addresses the second challenge: if after the perturbation, an AI model has low confidence in correctly predicting its target, then this perturbed data point likely behaves as an outlier relative to the model's training dataset. We discuss two key design aspects for achieving these goals:

---

[3]Empirically, this works well even when the actual data attribution method used by the `AI Developer` is more advanced ones, as shown in Section 6.2.

1. **Data component to be perturbed.** Recall that a data point $z = (x, y)$ consists of both the input feature $x$ and the prediction target $y$. To obtain an outlier data point from $z$, we perturb the feature $x$ only, without modifying the target $y$. Intuitively, flipping the target $y$ could easily result in an outlier data point through mislabeling, such an outlier typically degrades model performance and is likely to be identified as negatively influential. Moreover, mislabeled data points are easier for the `AI Developer` to detect. Therefore, it is more effective to perturb only the feature $x$ while keeping the target $y$ unchanged.

2. **Objective of perturbation.** For each data point $z = (x, y)$, we aim to decrease the model confidence in predicting the annotated target $y$ by perturbing $x$. In most cases, this is equivalent to increasing the loss $\ell$ between the model prediction and the target $y$.

Combining these two design choices, the resulting perturbed data points coincide with what is commonly referred to as *adversarial example* in the literature of adversarial attacks against neural network models. Consequently, existing adversarial attack methods for deriving adversarial examples can be leveraged to generate realistic outliers in our problem setup. Notably, this strategy is very general and can be applied to a variety of data modalities and models by leveraging different off-the-shelf black-box adversarial attack methods.

**Choices of Adversarial Attack Methods.** We consider two concrete machine learning settings, image classification, and text generation, and discuss the choices of adversarial attack methods. For smaller-scale image classification settings, we leverage Zeroth Order Optimization (ZOO) (Chen et al., 2017) based adversarial attacks, which approximates the gradient ascend on the loss function with respect to the data features using black-box queries to the target model $\mathcal{T}(Z_0)$. For larger-scale image classification settings, we employ a more advanced black-box adversarial attack method, Simba (Guo et al., 2019), which is computationally more efficient. At a high level, Simba sequentially perturbs each scalar pixel value in the image by trying to perturb in both directions and accept the perturbation once it increases the loss. For the text generation setting, we utilize TextFooler (Jin et al., 2020), a black-box adversarial attack method tailored for text data. In all the methods of choice, the attack only requires black-box queries to get the predictions of the target model.

## 5.3 Theoretical understanding

The following theorem further provides theoretical insights about the effectiveness of the proposed Outlier Attack. The formal statement, notations, and the proof can be found in Appendix A.

**Theorem 5.1** (Informal). *Consider a model trained by ERM on a dataset of size $n$ with a smooth loss $\ell$ with respect to model parameters $\theta$. Assume its corresponding influence score $\tau$, gradient $\nabla_\theta \ell(\theta, z)$, and Hessian $\nabla_\theta^2 \ell(\theta, z)$ are all bounded, i.e., $|\tau|, \|\nabla_\theta \ell\|_2, \|\nabla_\theta^2 \ell\|_{op} = \Theta_n(1)$. Assume the influence score $\tau$ is based on the* influence function *by Koh & Liang (2017), which takes the form*

$$\tau_{IF}(z_j, z_{test}; \hat{\theta}) = -\nabla_\theta \ell(\hat{\theta}, z_{test})^\top \left[ \frac{1}{n} \sum_{i=1}^n \nabla_\theta^2 \ell(\hat{\theta}, z_i) \right]^{-1} \nabla_\theta \ell(\hat{\theta}, z_j), \qquad (4)$$

*where $z_{test}$ is a test data point while $\{z_i\}_{i=1}^n$ are the training data points and $\hat{\theta}$ is the model parameters trained on $\{z_i\}_{i=1}^n$. Then, for any $z_{test}$, when $z_j$ in $\{z_i\}_{i=1}^n$ is perturbed to $z_j'$,*

- $\tau_{IF}(z_i, z_{test}; \hat{\theta}') = \tau_{IF}(z_i, z_{test}; \hat{\theta}) + O(1/n)$ *for all $i \neq j$, and*

- $\tau_{IF}(z_j', z_{test}; \hat{\theta}') = \tau_{IF}(z_j', z_{test}; \hat{\theta}) + O(1/n)$,

*where $\hat{\theta}'$ refers to the model parameters trained on the perturbed dataset $\{z_i\}_{i \neq j} \cup \{z_j'\}$.*

Intuitively, perturbing $z_j$ through adversarial attack will increase the magnitude of the gradient $\nabla_\theta \ell(\hat{\theta}, z_j)$, which tends to also increase the influence score on the original model $\hat{\theta}$, i.e., $\tau_{IF}(z_j', z_{test}; \hat{\theta}) \gg \tau_{IF}(z_j, z_{test}; \hat{\theta})$. However, in order to get a higher compensation share, the perturbed data point $z_j'$ needs to have a high influence score on the model $\hat{\theta}'$ trained on the perturbed dataset, i.e., $\tau_{IF}(z_j', z_{test}; \hat{\theta}')$, which is not guaranteed by the adversarial attack without characterizing the new model $\hat{\theta}'$. Theorem 5.1 exactly does this and asserts that $\tau_{IF}(z_j', z_{test}; \hat{\theta}')$ will be close

to $\tau_{\text{IF}}(z'_j, z_{\text{test}}; \hat{\theta})$ when the datasets used to train $\hat{\theta}$ and $\hat{\theta}'$ are close. It further guarantees that the influence scores of the rest unchanged data points will also remain similar. This explains why the effect of adversarial attacks on the original model can successfully translate to the new model. The results in Theorem 5.1 can be generalized to the case where more than one data point is perturbed, and a small set of clean data is also added to train the new model. The bound will change from $O(1/n)$ to $O(k/n)$, where $k$ is the total number of changed data points in the dataset.

## 6 EXPERIMENTS

In this section, we evaluate the effectiveness of the proposed attack methods by the increase of compensation share after manipulating the data with the proposed attack methods.

### 6.1 EXPERIMENTAL SETUP

**Tasks, Datasets, Target Models, and Data Attribution Methods.** We consider two machine-learning tasks: image classification and text generation. For image classification, we experiment on MNIST (LeCun, 1998), Digits (Jiang et al., 2023) and CIFAR-10 (Krizhevsky & Hinton, 2009) datasets, with different target models including Logistic Regression (LR), Multi-layer Perceptron (MLP), Convolutional Neural Networks (CNN), and ResNet-18 (He et al., 2016). We also employ three popular data attribution methods, including Influence Function (Koh & Liang, 2017), TRAK (Park et al., 2023), and Data Shapley (Ghorbani & Zou, 2019). For text generation, we conduct experiments on NanoGPT (Karpathy, 2022) trained on the Shakespeare dataset (Karpathy, 2015), with TRAK as the data attribution method. Finally, for the image classification settings, we evaluate both Shadow Attack and Outlier Attack, while for the text generation setting, we evaluate Outlier Attack only as Assumption 2 usually does not hold for generative AI settings. For the data attribution algorithms, we adopt the implementation from the dattri library (Deng et al., 2024). The experimental settings are summarized in Table 1.

Table 1: Summary of the experimental settings.

| Setting | Task | Dataset | Target Model | Attribution Method |
|---------|------|---------|--------------|--------------------|
| (a) | Image Classification | MNIST | LR | Influence Function |
| (b) | Image Classification | Digits | MLP | Data Shapley |
| (c) | Image Classification | MNIST | CNN | TRAK |
| (d) | Image Classification | CIFAR-10 | ResNet-18 | TRAK |
| (e) | Text Generation | Shakespeare | NanoGPT | TRAK |

**Data Contribution Workflow.** For the image classification settings (a), (c), and (d), we set $|Z_0| = 10000, |Z_1^a| = 100$, and $|Z_1| = 11000$. This setup simulates the following data contribution workflow: At $t = 0$ when there is no `Adversary`, and 10000 training points are used to train the model $\mathcal{T}(Z_0)$. At $t = 1$, the `Adversary` contributes 100 perturbed training points and other `Data Providers` contribute 900 new training data points. Together with the previous 10000 training points, a total of 11000 training points are used to train the model $\mathcal{T}(Z_1)$. For image classification setting (b), due to the size of the dataset, we set $|Z_0| = 1100, |Z_1^a| = 30$ and $|Z_1| = 1100$ for Outlier Attack, $|Z_0| = 800, |Z_1|^a = 30$ and $|Z_1| = 850$ for Shadow Attack. The text generation setting follows a similar workflow with $|Z_0| = 4706, |Z_1^a| = 20$, and $|Z_1| = 6274$.

**Evaluation Metrics.** We evaluate the proposed attack methods using two metrics, where we set $k = 100$ when counting the top-$k$ influential data points in both cases. The first is the **Compensation Share** (Eq. (1)) where for each attack method, we calculate $c(Z_1^a)$ respectively when the original dataset without perturbation is contributed as $Z_1^a$ (**Original**) and when the manipulated dataset after perturbation is contributed as $Z_1^a$ (**Manipulated**). The increase of $c(Z_1^a)$ after the perturbation measured by the **Ratio** reflects the effectiveness of the attack.

To gain a more refined understanding of how the adversarial perturbations affect the top-$k$ influential data points for individual validation data points in $V_1$, we consider the second evaluation metric named **Fraction of Change**. For each validation data point $v \in V_1$, it measures how many data points in $Z_1^a$ appear in the top-$k$ influential data points for $v$. In comparison to when the original

dataset without perturbation is contributed as $Z_1^a$, we calculate the fraction of validation data points in $V_1$ that contain more data points from $Z_1^a$ in the top-$k$ influential data points when the manipulated dataset after perturbation is contributed as $Z_1^a$. Similarly, we calculate the fraction of validation points that contain the same number of or fewer data points from $Z_1^a$ after perturbation. We report the three fractions under the categories **More**, **Tied**, and **Fewer**, where the higher fraction for the **More** category indicates that the attack method influences the validation data points more broadly.

## 6.2 EXPERIMENTAL RESULTS: SHADOW ATTACK

The results of the proposed Shadow Attack method are shown in Table 2. We conduct experiments on all three image classification settings outlined in Table 1. For each of the settings, we first consider the case that the shadow models have the same architecture as the target model, as shown in the first three rows of Table 2. Additionally, for setting (d) where the target model is ResNet-18 (forth row), we further consider using ResNet-9 as the shadow model (last row), which simulates scenarios where the `Adversary`'s knowledge about the training algorithm is limited.

Table 2: Results of the Shadow Attack method. The target models used for evaluation in each setting are listed in Table 1 while the shadow models used in the attack are listed under **Shadow Model**. The proportion $|Z_1^a|/|Z_1|$ of the data contributed by the `Adversary` relative to the full dataset at $t = 1$ is also reported. A higher **Ratio** indicates a more effective attack, and a higher fraction under **More** means that the attack influences the validation data points more broadly.

| Setting | Shadow Model | $|Z_1^a|/|Z_1|$ | Compensation Share | | | Fraction of Change | | |
|---------|-------------|-----------------|---------------------|---|---|--------------------|---|---|
| | | | Original | Manipulated | Ratio | More | Tied | Fewer |
| (a) | LR | 0.0098 | 0.0098 | 0.0477 | 456.1% | 0.955 | 0.038 | 0.007 |
| (b) | MLP | 0.0352 | 0.0152 | 0.0435 | 286.2% | 0.533 | 0.333 | 0.134 |
| (c) | CNN | 0.0098 | 0.0112 | 0.0467 | 417.0% | 0.781 | 0.195 | 0.024 |
| (d) | ResNet-18 | 0.0098 | 0.0095 | 0.0213 | 217.3% | 0.655 | 0.259 | 0.086 |
| (d) | ResNet-9 | 0.0098 | 0.0095 | 0.0196 | 206.3% | 0.622 | 0.310 | 0.068 |

The Shadow Attack method is highly effective in increasing the **Compensation Share** across all the settings. The **Ratio** of the **Manipulated** to the **Original** ranges from $206.3\%$ to $456.1\%$, representing a substantial increase. Notably, the last row corresponds to the setup where the shadow models have the ResNet-9 architecture while the target model is a ResNet-18, and the Shadow Attack remains significantly effective ($206.3\%$,) although being slightly worse than the case where shadow model and target model shares the same architecture (forth row, $217.3\%$.).

The Shadow Attack is also uniformly effective across a wide range of validation data points, as measured by the metrics of **Fraction of Change**. The attack is able to increase the number of top-$k$ influential points from $Z_1^a$ for more than $60\%$ of the validation points. In contrast, only less than $9\%$ of the validation data points have fewer top-$k$ influential points from $Z_1^a$ after the attack.

## 6.3 EXPERIMENTAL RESULTS: OUTLIER ATTACK

The results of the proposed Outlier Attack method are presented in Table 3, where we include all four settings summarized in Table 1. The black-box attack method for generating the adversarial examples is chosen according to the discussion in Section 5.2.

On experimental settings (a) and (c), the Outlier Attack performs even better than the Shadow Attack in terms of both **Compensation Share** and **Fraction of Change**, achieving a higher **Ratio** and a larger fraction under **More**. Across all four settings, the **Ratio** ranges from $185.2\%$ to $643.9\%$, demonstrating the exceptional effectiveness of the Outlier Attack. Finally, the results of text generation setting (e) further highlight the applicability of the proposed method to generative AI models.

## 6.4 BASELINE REFERENCE

To better understand the significance of the results in Table 2 and Table 3, we compare them with a baseline random perturbation method in Table 4 on the three image classification settings such that the baseline method applies pixel-wise random perturbation to the images, with the perturbation

Table 3: Results of the Outlier Attack method. The black-box adversarial attack methods for generating adversarial examples are listed under **Attack Method**. See Table 2 for more context.

| Setting | Attack Method | $|Z_1^a|/|Z_1|$ | Compensation Share | | | Fraction of Change | | |
|---------|---------------|-----------------|----------|------------|-------|------|------|-------|
| | | | Original | Manipulated | Ratio | More | Tied | Fewer |
| (a) | ZOO | 0.0098 | 0.0098 | 0.0631 | 643.9% | 0.980 | 0.017 | 0.003 |
| (b) | Simba | 0.0250 | 0.0112 | 0.0218 | 194.6% | 0.440 | 0.380 | 0.180 |
| (c) | Simba | 0.0098 | 0.0112 | 0.0668 | 596.4% | 0.799 | 0.173 | 0.028 |
| (d) | Simba | 0.0098 | 0.0095 | 0.0176 | 185.2% | 0.562 | 0.354 | 0.084 |
| (e) | TextFooler | 0.0013 | 0.0035 | 0.0092 | 262.9% | 0.392 | 0.461 | 0.147 |

budget matched to that of the proposed attack methods. As shown in the results, the **Compensation Share** of the dataset with **Random Perturbation** is nearly identical to that of the **Original**. The fraction under **More** is also close to that under **Fewer**. These results highlight the effectiveness of the proposed attack methods comes from the careful design of adversarial perturbation.

Table 4: Results of the random perturbation baseline. See Table 2 for more context.

| Setting | Compensation Share | | Fraction of Change | | |
|---------|----------|---------------------|------|------|-------|
| | Original | Random Perturbation | More | Tied | Fewer |
| (a) | 0.0098 | 0.0097 | 0.203 | 0.586 | 0.211 |
| (c) | 0.0112 | 0.0117 | 0.385 | 0.326 | 0.289 |
| (d) | 0.0095 | 0.0125 | 0.367 | 0.430 | 0.203 |

## 7 CONCLUSION

This work addresses a significant gap in the current understanding of the adversarial robustness of data attribution methods, which are increasingly influential in data valuation and compensation applications. By introducing a well-defined threat model, we have proposed two novel adversarial attack strategies, Shadow Attack and Outlier Attack, which are designed to manipulate data attribution and inflate compensation. The Shadow Attack utilizes knowledge of the underlying data distribution, while the Outlier Attack operates without such knowledge, relying on black-box queries only. Empirical results from image classification and text generation tasks demonstrate the effectiveness of these attacks, with compensation inflation ranging from 185% to 643%. These findings underscore the need for more robust data attribution methods to guard against adversarial exploitation.

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

# A    FORMAL STATEMENT OF THEOREM 5.1 AND ITS PROOF

## A.1    SETUP AND THE STATEMENT

Let the original training set be $Z = \{z_i\}_{i=1}^n$. Given a data point $z$, assume the loss $\ell(\theta, z)$ is twice differentiable. Then, the empirical loss and the Hessian is given by

$$\ell(\theta) = \frac{1}{n} \sum_{i=1}^n \ell(\theta, z_i), \quad H(\theta) = \frac{1}{n} \sum_{i=1}^n h(\theta, z_i),$$

where we define $h(\theta, z) = \nabla_\theta^2 \ell(\theta, z)$. Moreover, we consider the ERM to be

$$\hat{\theta} = \arg\min_\theta \frac{1}{n} \sum_{i=1}^n \ell(\theta, z_i)$$

Consider the training data attribution to be the influence estimated by the *influence function* (Koh & Liang, 2017) $\tau_{\text{IF}}$. In particular, for a train test pair $(z_i, z_{\text{test}})$, Eq.(4) can be succinctly written as

$$\tau_{\text{IF}}(z_i, z_{\text{test}}; \hat{\theta}) = -\nabla_\theta \ell(\hat{\theta}, z_i)^\top H^{-1}(\hat{\theta}) \nabla_\theta \ell(\hat{\theta}, z_{\text{test}}).$$

Consider perturbing the $j^{\text{th}}$ training point such that $z_j$ becomes $z_j'$. Then, the training set turns to $\{z_1, \ldots, z_j', \ldots, z_n\}$, which in turn changes the empirical loss and its hessian to

$$\ell'(\theta) = \frac{1}{n} \sum_{i \neq j} \ell(\theta, z_i) + \frac{1}{n} \ell(\theta, z_j'), \quad H'(\theta) = \frac{1}{n} \sum_{i \neq j} h(\theta, z_i) + \frac{1}{n} h(\theta, z_j').$$

It also follows that the minimizer for this perturbed training set is given by

$$\hat{\theta}' = \arg\min_\theta \frac{1}{n} \sum_{i \neq j} \ell(\theta, z_i) + \frac{1}{n} \ell(\theta, z_j'),$$

with the corresponding influence score being

- $\tau_{\text{IF}}(z_i, z_{\text{test}}; \hat{\theta}') = -\nabla_\theta \ell(\hat{\theta}', z_{\text{test}})^\top (H'(\hat{\theta}'))^{-1} \nabla_\theta \ell(\hat{\theta}', z_i)$ for all $i \neq j$; and
- $\tau_{\text{IF}}(z_j', z_{\text{test}}; \hat{\theta}') = -\nabla_\theta \ell(\hat{\theta}', z_{\text{test}})^\top (H'(\hat{\theta}'))^{-1} \nabla_\theta \ell(\hat{\theta}', z_j').$

**Remark 1.** *From our definition of data attribution methods, $\tau$ should take the form of $Z \times V \to \mathbb{R}$ for some training dataset $Z$ and validation dataset $V$. In the above notation, we are essentially considering two different training attribution functions $\tau_{IF}$ and $\tau_{IF}'$ where $\tau_{IF} = \mathcal{A}(Z, \mathcal{T}(Z), V)$ and $\tau_{IF}' = \mathcal{A}(Z', \mathcal{T}(Z'), V)$ where $Z'$ is the perturbed dataset, $\hat{\theta} = \mathcal{T}(Z)$ and $\hat{\theta}' = \mathcal{T}(Z')$. For clarity, we explicitly write $\tau_{IF}(\cdot, \cdot) = \tau_{IF}(\cdot, \cdot; \hat{\theta})$ and $\tau_{IF}'(\cdot, \cdot) = \tau_{IF}(\cdot, \cdot; \hat{\theta}')$ to avoid confusion.*

Under this setup, we can now state the theorem formally.

**Theorem A.1.** *Under the above setup, consider a model trained by ERM with a loss $\ell$ that is twice-differentiable, $m$-strongly convex, and $L$-Lipschitz continuous with respect to $\theta$. Assume its corresponding influence score $\tau_{IF}$, gradient $\nabla_\theta \ell(\theta, z)$, and $h(\theta, z)$ are all bounded, i.e., $|\tau_{IF}| = \Theta(1)$, $\|\nabla_\theta \ell\|_2 = \Theta(1)$, $\|h\|_{\text{op}} = \Theta(1)$. Then, given any test data point $z_{test}$, we have*

- $\tau_{IF}(z_i, z_{test}; \hat{\theta}') = \tau_{IF}(z_i, z_{test}; \hat{\theta}) + O(1/n)$ *for all $i \neq j$, and*
- $\tau_{IF}(z_j', z_{test}; \hat{\theta}') = \tau_{IF}(z_j', z_{test}; \hat{\theta}) + O(1/n).$

## A.2    TECHNICAL LEMMAS

We first establish several technical lemmas toward proving Theorem A.1.

**Lemma 1.** *Let $\hat{\theta}, \hat{\theta}'$ be the minimizer for $\ell(\theta), \ell'(\theta)$ respectively, then*

$$\|\hat{\theta} - \hat{\theta}'\| \leq \frac{4L}{mn}.$$

*Proof.* Firstly, we recall that from definition, $f(x)$ being $m$-strongly convex means that for any $x, y$,

$$f(y) \geq f(x) + \nabla f(x)^\top (y - x) + \frac{m}{2} \|y - x\|_2^2.$$

Denote $x^\star = \arg\min_x f(x)$, then since $\nabla f(x^\star) = 0$, for all $x$, we have

$$f(x) \geq f(x^\star) + \frac{m}{2} \|x - x^\star\|_2^2.$$

Hence, by strong convexity of $\ell(\theta, z)$ and the fact that $\hat{\theta}$ is the minimizer of $\ell(\theta)$, we have

$$\frac{1}{n} \sum_{i=1}^n \ell(\hat{\theta}', z_i) \geq \frac{1}{n} \sum_{i=1}^n \ell(\hat{\theta}, z_i) + \frac{m}{2} \|\hat{\theta} - \hat{\theta}'\|_2^2.$$

On the other hand, we have

$$\frac{1}{n} \sum_{i=1}^n \ell(\hat{\theta}', z_i) = \underbrace{\frac{1}{n} \sum_{i \neq j} \ell(\hat{\theta}', z_i) + \frac{1}{n} \ell(\hat{\theta}', z_j')}_{\ell'(\hat{\theta}')} + \frac{1}{n} (\ell(\hat{\theta}', z_j) - \ell(\hat{\theta}', z_j'))$$

$$\leq \underbrace{\frac{1}{n} \sum_{i \neq j} \ell(\hat{\theta}, z_i) + \frac{1}{n} \ell(\hat{\theta}, z_j')}_{\ell'(\hat{\theta})} + \frac{1}{n} (\ell(\hat{\theta}', z_j) - \ell(\hat{\theta}', z_j')) \quad (\hat{\theta}' \text{ is a minimizer of } \ell')$$

$$= \frac{1}{n} \sum_{i=1}^n \ell(\hat{\theta}, z_i) + \frac{1}{n} (\ell(\hat{\theta}', z_j) - \ell(\hat{\theta}, z_j)) + \frac{1}{n} (\ell(\hat{\theta}, z_j') - \ell(\hat{\theta}', z_j'))$$

$$\leq \frac{1}{n} \sum_{i=1}^n \ell(\hat{\theta}, z_i) + \frac{2}{n} L \|\hat{\theta} - \hat{\theta}'\|_2. \quad\quad (\ell(\theta, z) \text{ is } L\text{-Lipschitz w.r.t. } \theta)$$

Combine the above results, we see that

$$\frac{m}{2} \|\hat{\theta} - \hat{\theta}'\|_2^2 \leq \frac{2L}{n} \|\hat{\theta} - \hat{\theta}'\|_2,$$

which proves the desired result. $\square$

**Lemma 2** (Section 2.4, Part III (Stewart & Sun, 1990))**.** *For any $A, E \in \mathbb{R}^{d \times d}$ with both $A$ and $A + E$ being invertible, if $\sum_{k=0}^\infty \|E\|_{op}^k$ converges, we have*

$$(A + E)^{-1} = A^{-1} - A^{-1} E A^{-1} + O(\|E\|_{op}^2).$$

**Lemma 3.** *Let $H$ and $H'$ be the Hessian of $\ell$ and $\ell'$, respectively, then*

$$\|H'^{-1} - H^{-1}\|_{op} \leq O \left( \frac{4LM}{m^3 n} \right)$$

*Proof.* Since $h(\theta, z)$ is $M$-Lipschitz w.r.t. $\theta$, we know that

$$\|H' - H\|_{op} \leq M \|\hat{\theta}' - \hat{\theta}\|_2 \leq \frac{4LM}{mn}.$$

With Lemma 2 (let $A = H$ and $E = H' - H$, it's easy to verify the conditions of Lemma 2 hold), we have $H'^{-1} = H^{-1} - H^{-1}(H' - H)H^{-1} + O(1/n^2)$, which gives

$$\|H'^{-1} - H^{-1}\|_{op} \leq \|H^{-1}\|_{op} \|H' - H\|_{op} \|H^{-1}\|_{op} + O(1/n^2).$$

Since $\ell(\theta, z)$ is $m$-strongly convex w.r.t. $\theta$, it's easy to show that $\|H^{-1}\|_{op} \leq 1/m$. In all, we have

$$\|H'^{-1} - H^{-1}\|_{op} = O \left( \frac{4LM}{m^3 n} \right).$$

$\square$

## A.3 PROOF OF THEOREM A.1

We can now prove Theorem A.1.

*Proof of Theorem A.1.* For all $i \neq j$, we want to prove that $\tau_{\text{IF}}(z_i, z_{\text{test}}; \hat{\theta}') = \tau_{\text{IF}}(z_i, z_{\text{test}}; \hat{\theta}) + O(1/n)$. Indeed, since

$$
\begin{aligned}
\tau_{\text{IF}}(z_i, z_{\text{test}}; \hat{\theta}') &= -\nabla_\theta \ell(\hat{\theta}', z_{\text{test}})^\top (H'(\hat{\theta}'))^{-1} \nabla_\theta \ell(\hat{\theta}', z_i) \\
&= \big(\nabla_\theta \ell(\hat{\theta}, z_{\text{test}}) - \nabla_\theta \ell(\hat{\theta}', z_{\text{test}})\big)^\top (H'(\hat{\theta}'))^{-1} \nabla_\theta \ell(\hat{\theta}, z_i) \\
&\quad - \nabla_\theta \ell(\hat{\theta}, z_{\text{test}}) \big((H'(\hat{\theta}'))^{-1} - H^{-1}(\hat{\theta})\big) \nabla_\theta \ell(\hat{\theta}, z_i) \\
&\quad - \nabla_\theta \ell(\hat{\theta}, z_{\text{test}}) H^{-1}(\hat{\theta}) \nabla_\theta \ell(\hat{\theta}, z_i) \\
&= \big(\nabla_\theta \ell(\hat{\theta}, z_{\text{test}}) - \nabla_\theta \ell(\hat{\theta}', z_{\text{test}})\big)^\top (H'(\hat{\theta}'))^{-1} \nabla_\theta \ell(\hat{\theta}, z_i) \\
&\quad - \nabla_\theta \ell(\hat{\theta}, z_{\text{test}}) \big((H'(\hat{\theta}'))^{-1} - H^{-1}(\hat{\theta})\big) \nabla_\theta \ell(\hat{\theta}, z_i) \\
&\quad + \tau_{\text{IF}}(z_i, z_{\text{test}}; \hat{\theta}).
\end{aligned}
$$

Applying Lemmas 1 and 3, and by noting that when the Hessian is $M$-Lipschitz, so is the gradient in terms of $\theta$, hence we have the desired result. Specifically, we have

$$
\begin{aligned}
&|\tau_{\text{IF}}(z_i, z_{\text{test}}; \hat{\theta}') - \tau_{\text{IF}}(z_i, z_{\text{test}}; \hat{\theta})| \\
&\leq \|\nabla_\theta \ell(\hat{\theta}, z_{\text{test}}) - \nabla_\theta \ell(\hat{\theta}', z_{\text{test}})\|_2 \|H'^{-1}\|_{\text{op}} \|\nabla_\theta \ell(\hat{\theta}, z_i)\|_2 \\
&\quad + \|\nabla_\theta \ell(\hat{\theta}, z_{\text{test}})\|_2 \|H'^{-1} - H^{-1}\|_{\text{op}} \|\nabla_\theta \ell(\hat{\theta}, z_i)\|_2 \\
&\leq \|\nabla_\theta \ell(\hat{\theta}, z_i)\|_2 \big(\|H^{-1}\|_{\text{op}} + \|H'^{-1} - H^{-1}\|_{\text{op}}\big) \cdot M \|\hat{\theta}' - \hat{\theta}\|_2 \\
&\quad + \|\nabla_\theta \ell(\hat{\theta}, z_{\text{test}})\|_2 \|H'^{-1} - H^{-1}\|_{\text{op}} \|\nabla_\theta \ell(\hat{\theta}, z_i)\|_2 \\
&= \Theta(1) \cdot \left(\frac{1}{m} + O\left(\frac{4LM}{m^3 n}\right)\right) \cdot M \cdot \frac{4L}{mn} + \Theta(1) \cdot O\left(\frac{4LM}{m^3 n}\right) \cdot \Theta(1) = O\left(\frac{1}{n}\right).
\end{aligned}
$$

For the second case, the proof is the same by replacing $z_i$ with $z'_j$ in the above calculation. $\square$

## B EXPERIMENT DETAILS

In this section, we introduce the experiment details.

### B.1 DATASETS AND MODELS

**Logistic Regression and Convolutional Neural Network on MNIST.** For experiments on MNIST, we consider two different target models, Logistic Regression (LR) and convolutional neural network (CNN). The CNN comprises two convolutional layers: the first with 32 filters and the second with 64 filters (size $3 \times 3$, stride 1, padding 1), both followed by ReLU and max-pooling layers. The output is flattened into a vector of size $64 \times 7 \times 7$, which is then passed through a fully connected layer with 128 units before reaching the output layer with 10 classes. Both target models are trained on the first 10000 training points of the MNIST dataset. For shadow models, we train 50 of them, each on a randomly sampled subset of 5000 among the second 10000 training data points to ensure no overlaps with the first 10000 training data points used for target model training.

In both the target model training and shadow model training, we train LR for 30 epochs with SGD and a learning rate of 0.01. We train CNN for 50 epochs with Adam and a learning rate of 0.001.

**MLP on Digits.** For experiments on Digits, we consider the target MLP model with 5 hidden layers, each has 10 hidden neurons. Due to the size limit of the Digits dataset, we train the target model for an Outlier Attack using the first 1100 training points, while for the shadow attack, we train the target model on the first 800 data points and the shadow models on the second 800 data points. In all training, we again train the MLP for 30 epochs using Adam with an initial learning rate of 0.001.

**ResNet18 on CIFAR-10.** For experiments on CIFAR-10, we consider the classical ResNet18 model without dropouts. We use 10000 training points for training the target model. For shadow model training, we consider the classical ResNet-9 and ResNet-18 as the shadow models separately, again without dropouts. We train 50 shadow models, and each on a subset of 10000 training points that are disjoint with the training set of the target model. In all training, each model is trained for 100 epochs using Adam, with a learning rate of 0.001.

**NanoGPT on Shakespeare.** For experiments on the Shakespeare dataset, we consider the NanoGPT model, which is a character-level GPT model with 4 layers, 4 heads, and 128 dimension of the embedding. The block size is 64, and the batch size is 32. For both training at $t = 0$ and $t = 1$, we train the model for 2000 epochs, both using Adam with a learning rate of $6 \times 10^{-4}$. Note that since we only consider Outlier Attack for this setup, hence there is no shadow model training involved.

## B.2 TRAINING DATA ATTRIBUTION METHODS

In this section, we briefly introduce the attribution methods that are included in the evaluation. Given a training dataset $\{z_i\}_{i=1}^n$, we are interested in the data attribution of a particular training data point $z_j$ and a test data point $z_{\text{test}}$.

**Influence Function based on the Conjugate Gradients.** As we have seen, the original definition of *influence function* (Koh & Liang, 2017) is given by Eq.(4), i.e.,

$$\tau_{\text{IF}}(z_j, z_{\text{test}}; \hat{\theta}) = -\nabla_\theta \ell(\hat{\theta}, z_j)^\top H(\hat{\theta})^{-1} \nabla_\theta \ell(\hat{\theta}, z_{\text{test}})$$

where $\nabla_\theta \ell(\hat{\theta}, z)$ is the gradient of loss of the data point w.r.t. model parameters, and $H(\hat{\theta})^{-1}$ is the inverse of the Hessian w.r.t. model parameters. We implement the conjugate gradients (CG) approach to compute the inverse-Hessian-vector-product (IHVP).

**Tracing with the Randomly-projected After Kernel (TRAK).** Introduced by Park et al. (2023), *TRAK* computes the attribution score for a training test pair by first linearizing the model and then applying random projection to make the computation efficient. Denote the output (e.g., raw logit) of the model $\hat{\theta}$ to be $f(z; \hat{\theta})$, then the *naive TRAK influence* can be formulated as

$$\tau_{\text{TRAK}}(z_j, z_{\text{test}}; \hat{\theta}) = -(1 - p_j^\star)\phi(z_j)^\top \left(\Phi^\top \Phi\right)^{-1} \phi(z_{\text{test}}),$$

where $\phi(z) = P^\top \nabla_\theta f(z; \hat{\theta})$ is the random projection of $\nabla_\theta f(z; \hat{\theta}) \in \mathbb{R}^p$ by some Gaussian random projection matrix $P \sim \mathcal{N}(0, 1)^{p \times k}$, and $\Phi$ is the matrix formed by stacking all the $\phi(z_i)$, and $p_j^\star$ is the predicted correct-class probability of $z_j$ at $\hat{\theta}$. Compare it with $\tau_{\text{IF}}$:

1. The additional factor $1 - p_j^\star$ in $\tau_{\text{TRAK}}$ is due to linearizing the model.

2. For linear model with feature matrix $X$ of the training set, its Hessian is exactly $X^\top X$. For for a linearized the model, each feature $x_i$ of training sample $z_i$ corresponds to the gradient $\nabla_\theta f(z_i; \hat{\theta})$, i.e., $X = [\nabla_\theta f(z_i; \hat{\theta})]_{i=1}^n$.

3. With random projection, the linearized features $x_j = \nabla_\theta f(z_i; \hat{\theta})$ becomes $\phi(z_i) = P\nabla_\theta f(z_i; \hat{\theta})$, which induces a new feature matrix $\Phi = [\phi(z_i)]_{i=1}^n$

Hence, overall, $\tau_{\text{TRAK}}$ is nothing but the influence function applied to the linearized model with random projection. We note that the original TRAK influence includes a step called *ensembling*, which is simply averaging the above TRAK influence over multiple $\tau_{\text{TRAK}}$ with models independently trained on a subset of the training set.

**Grad-Dot.** Introduced by Charpiat et al. (2019), the dot product of gradient, known as *Grad-Dot*, is a simple, easy-to-compute training data attribution method. It is given by

$$\tau_{\text{Grad-Dot}}(z_j, z_{\text{test}}; \hat{\theta}) = -\nabla_\theta \ell(\hat{\theta}, z_j)^\top \nabla_\theta \ell(\hat{\theta}, z_{\text{test}}).$$

**Data Shapley.** *Data Shapley* (Ghorbani & Zou, 2019) quantifies the influence of individual data points by considering its marginal contribution to different subsets of the training set. It is motivated by the so-called leave-one-out (LOO) influence: specifically, given any aspect $\phi(\hat{\theta})$ we care about for the learned model $\hat{\theta}$ (e.g., the loss $\ell(\hat{\theta}, z_{\text{test}})$ of some test data point $z_{\text{test}}$), LOO measures the influence of every individual data point $z_j$ on $\phi$ by the difference $\phi(\hat{\theta}_{-z_j}) - \phi(\hat{\theta})$, where $\hat{\theta}_{-z_j}$ is the model learned with the dataset $\{z_i\}_{i \neq j}$ that excludes $z_j$. Data Shapley builds on top of LOO by requiring additional *equitable* conditions, resulting in the following formulation

$$\tau_{\text{Data-Shapley}}(z_j, z_{\text{test}}; \hat{\theta}) = \sum_{S \subseteq \{z_i\}_{i \neq j}} \frac{\phi(\hat{\theta}_S, z_{\text{test}}) - \phi(\hat{\theta}_{S \cup \{j\}}, z_{\text{test}})}{\binom{n-1}{|S|}},$$

where we let $\phi(\hat{\theta}, z_{\text{test}}) = \ell(\hat{\theta}, z_{\text{test}})$ and $\hat{\theta}_S$ for some $S \subseteq \{z_i\}_{i=1}^{n} \setminus \{z_j\}$ denotes the model learned on the subset $S$. This can be viewed as an averaged version of LOO while satisfying several equitable conditions, which we refer to Ghorbani & Zou (2019) for details.

**Remark 2.** *We note that in the above formulations, we incorporate a negative sign in front of $\tau_{TRAK}$, $\tau_{Grad\text{-}Dot}$ to make them consistent to the original formulation of $\tau_{IF}$. Specifically, in Koh & Liang (2017), the influence change is measured as the difference between the perturbed model and the original model, while others consider the opposite.*

## B.3 ATTACK METHODS

In this section, we detail the implementation of all the attack methods we have used throughout the experiments.

**Shadow Attack.** Shadow attack is a classical adversarial attack method that originates from (Shokri et al., 2017). The details of shadow model training can be found in Appendix B.1. As introduced in Section 4, after 50 shadow models are trained, we optimize Eq. (3) using gradient ascent with respect to input feature $x$ with 10 iterations and a step size of $\epsilon = 0.01$.

**Outlier Attack.** For Outlier Attack, we utilize the following black-box attack methods to produce adversarial examples:

- **Zeroth Order Optimization (ZOO)**: ZOO is a black-box attack method that approximates gradient through finite numerical methods. In our experiment, given an input $z = (x, y)$, we perturb it by $x' \leftarrow x + \epsilon \cdot \text{sgn}(g(\hat{\theta}, z))$, where $g(\hat{\theta}, z)$ is an estimation of the loss of the gradient with respect to $x$, given by the symmetric difference quotient

  $$\left(g(\hat{\theta}, z)\right)_i = \frac{\ell(\hat{\theta}, (x + h\mathbf{e}_i, y)) - \ell(\hat{\theta}, (x - h\mathbf{e}_i, y))}{2h},$$

  along the $i^{\text{th}}$ standard basis direction $\mathbf{e}_i$ with $h \in \mathbb{R}$. We note that $\ell(\hat{\theta}, (x, y))$ can be obtained by black-box queries of the target model. In experiments, we set $\epsilon = 0.03$.

- **Simba**: Introduced by Guo et al. (2019), Simba proposes a black-box attack method to perturb each input pixel sequentially after permutation. More precisely, each input pixel is attempted to be perturbed in both directions, i.e., $(x')_i \leftarrow x_i \pm \epsilon$, respectively. After a perturbation is attempted, the target model is queried again, and the perturbation is accepted as long as this perturbation increases the loss. In practice, we set $\epsilon = 0.1$.

- **TextFooler**: TextFooler is originally an attack method for text *classification* task (Jin et al., 2020). It perturbs texts following a two-step approach: for a piece of text that it wants to perturb, it first sorts the words by their importance, and then replaces the influential words to increase the loss of prediction while preserving text similarity.

  We modify the method as follows. Firstly, for a character-level GPT model (in our case, NanoGPT), we work on characters rather than words. Secondly, as we work on text *generation* task, for a sequence of characters that we want to perturb, we take the sum of the negative log-likelihood of predicting its $m$ future characters as a loss. We then sort the characters by their importance, where importance is measured by the loss increase after the character is masked. Finally, the $k$-most important characters are replaced by characters that maximize the increment of the loss. In experiments we set $m = 20$ and $k = 15$.

# C  ADDITIONAL EXPERIMENTS

In this section, we conduct additional experiments under various settings, including ablation studies and more.

## C.1  ABLATION STUDY: CHANGING VALUE OF $|Z_0|$ AND $|Z_0!|$

We test the performance of our methods under different sizes of $|Z_0|$ and $|Z_1|$ to understand the effect of data set sizes. For settings (a), (c), and (d) in Table 1, we consider decreasing $(|Z_0|, |Z_1|)$ from $(10000, 11000)$ to $(5000, 6000)$ and increasing $(|Z_0|, |Z_1|)$ from $(10000, 11000)$ to $(15000, 16000)$, experimenting with both the Shadow Attack and the Outlier Attack. For the setting (e), i.e., the text generation setup, the original $|Z_0|$ was already small (originally, $(|Z_0|, |Z_1|) = (4706, 6274)$), which takes only $30\%$ of the data. Hence, we consider increasing the data size to two different extents. Note that since the Shadow Attack is infeasible in this setting, only the Outlier Attack's results are shown for setting (2). The results for the two attack methods are respectively shown in Tables 5 and 6. We see that the results demonstrate the effectiveness of our methods across different sizes of the dataset. Overall, with a few exceptions, the **Ratio** of the **Compensation Share** appears to become even higher when the dataset size is larger.

Table 5: Results of Shadow Attack for various $(|Z_0|, |Z_1|)$ settings.

| Setting | $(|Z_0|, |Z_1|)$ | Compensation Share | | | Fraction of Change | | |
|---|---|---|---|---|---|---|---|
| | | Original | Manipulated | Ratio | More | Tied | Fewer |
| (a) | (5000, 6000) | 0.0153 | 0.0764 | 499.3% | 0.996 | 0.003 | 0.001 |
| | (10000, 11000) | 0.0098 | 0.0477 | 456.1% | 0.955 | 0.038 | 0.007 |
| | (15000, 16000) | 0.0050 | 0.0373 | 746.0% | 0.963 | 0.036 | 0.001 |
| (c) | (5000, 6000) | 0.0168 | 0.0539 | 320.7% | 0.857 | 0.129 | 0.014 |
| | (10000, 11000) | 0.0112 | 0.0467 | 417.0% | 0.781 | 0.195 | 0.024 |
| | (15000, 16000) | 0.0002 | 0.0062 | 3100.0% | 0.431 | 0.568 | 0.001 |
| (d) | (5000, 6000) | 0.0206 | 0.0413 | 200.7% | 0.696 | 0.174 | 0.130 |
| | (10000, 11000) | 0.0095 | 0.0213 | 217.3% | 0.655 | 0.259 | 0.086 |
| | (15000, 16000) | 0.0057 | 0.0092 | 161.5% | 0.264 | 0.616 | 0.120 |

Table 6: Results of Outlier Attack for various $(|Z_0|, |Z_1|)$ settings.

| Setting | $(|Z_0|, |Z_1|)$ | Compensation Share | | | Fraction of Change | | |
|---|---|---|---|---|---|---|---|
| | | Original | Manipulated | Ratio | More | Tied | Fewer |
| (a) | (5000, 6000) | 0.0153 | 0.0747 | 488.9% | 0.998 | 0.002 | 0.000 |
| | (10000, 11000) | 0.0098 | 0.0631 | 643.9% | 0.980 | 0.017 | 0.003 |
| | (15000, 16000) | 0.0050 | 0.0400 | 800.0% | 0.964 | 0.036 | 0.000 |
| (c) | (5000, 6000) | 0.0168 | 0.0334 | 198.8% | 0.720 | 0.241 | 0.039 |
| | (10000, 11000) | 0.0112 | 0.0668 | 596.4% | 0.799 | 0.173 | 0.028 |
| | (15000, 16000) | 0.0002 | 0.0051 | 2550.0% | 0.397 | 0.603 | 0.000 |
| (d) | (5000, 6000) | 0.0206 | 0.0411 | 199.5% | 0.761 | 0.133 | 0.106 |
| | (10000, 11000) | 0.0095 | 0.0176 | 185.2% | 0.562 | 0.354 | 0.084 |
| | (15000, 16000) | 0.0057 | 0.0219 | 384.2% | 0.731 | 0.192 | 0.077 |
| (e) | (4706, 6274) | 0.0031 | 0.0035 | 262.9% | 0.392 | 0.461 | 0.147 |
| | (7843, 9411) | 0.0016 | 0.0064 | 400.0% | 0.420 | 0.507 | 0.073 |
| | (12549, 14116) | 0.0029 | 0.0214 | 737.9% | 0.400 | 0.543 | 0.057 |

## C.2  ABLATION STUDY: DATA ATTRIBUTION METHOD

Next, we conduct an ablation study by varying the data attribution methods used in the evaluation under the experiment setting (b) in Table 1. Specifically, apart from Data Shapley, we consider

two additional data attribution methods, Influence Function, and TRAK, for evaluating the compensation share, experimenting with both the Shadow Attack and the Outlier Attack. Note that the `Adversary` has no knowledge about what data attribution method will be used by the `AI Developer`. The results are shown in Tables 7 and 8. The results below show that the proposed attack methods are highly effective for all three data attribution methods, as reflected by the **Ratio** of the **Compensation Share**.

Table 7: Results of Shadow Attack with different data attribution methods of setting (b).

| Attribution Method | $|Z_1^a|/|Z_1|$ | Compensation Share | | | Fraction of Change | | |
|---|---|---|---|---|---|---|---|
| | | Original | Manipulated | Ratio | More | Tied | Fewer |
| Data Shapley | 0.0352 | 0.0152 | 0.0435 | 286.2% | 0.533 | 0.333 | 0.134 |
| Influence Function | 0.0352 | 0.0496 | 0.1004 | 202.4% | 0.980 | 0.000 | 0.020 |
| TRAK | 0.0352 | 0.0392 | 0.0936 | 238.8% | 0.820 | 0.100 | 0.080 |

Table 8: Results of Outlier Attack with different data attribution methods of setting (b).

| Attribution Method | $|Z_1^a|/|Z_1|$ | Compensation Share | | | Fraction of Change | | |
|---|---|---|---|---|---|---|---|
| | | Original | Manipulated | Ratio | More | Tied | Fewer |
| Data Shapley | 0.0250 | 0.0112 | 0.0218 | 194.6% | 0.440 | 0.380 | 0.180 |
| Influence Function | 0.0250 | 0.0186 | 0.0442 | 237.6% | 0.920 | 0.060 | 0.020 |
| TRAK | 0.0250 | 0.0160 | 0.0412 | 257.5% | 0.780 | 0.180 | 0.040 |

## C.3 WHITE-BOX ATTACK

In this section, we further consider *white-box* attacks as an oracle reference, where the `Adversary` has full knowledge of the target model and has access to the model parameters. For the choice of attack method under the white-box threat model, we experiment with the Fast Gradient Sign Method (FGSM) (Goodfellow et al., 2015) for setting (a) and the Projected Gradient Descent(PGD) (Mkadry et al., 2017) for setting (d). The results are shown in Table 9. Overall, compared to this oracle white-box attack, the proposed black-box attacks are only slightly worse in terms of the **Ratio** of the **Compensation Share**. This further confirms that the proposed methods are highly effective.

Table 9: Results of white-box attacks under settings (a) and (d).

| Setting | Attack Method | Compensation Share | | | Fraction of Change | | |
|---|---|---|---|---|---|---|---|
| | | Original | Manipulated | Ratio | More | Tied | Fewer |
| (a) | FGSM | 0.0098 | 0.0649 | 662.2% | 0.990 | 0.007 | 0.003 |
| (d) | PGD | 0.0095 | 0.0222 | 233.7% | 0.689 | 0.204 | 0.107 |

## C.4 ADVERSARIAL ATTACK UNDER DATA AUGMENTATION

In this section, we test the performance of our proposed attack methods when the `AI Developer` utilize data augmentation techniques when training the model. Specifically, after the `AI Developer` gathers data, the training dataset is then formed by a combination of the gathered data and its augmented version. For simplicity, we consider settings (a), (c), and (d) in Table 1 and consider standard image augmentation methods such as random cropping and random affine transformation to images. With data augmentation, the compensation share will be attributed back to the **original** data point if its augmented version is identified as influential. The results are shown in Tables 10 and 11. Compared to the original setting without data augmentation, we cannot draw a definite conclusion on whether data augmentation helps defend the proposed attacks since the trend is unclear. However, overall, we can conclude that the proposed attacks are still highly effective even when data augmentation is utilized by the `AI Developer`.

Table 10: Results of Shadow Attack under data augmentation

| Setting | $|Z_1^a|/|Z_1|$ | Compensation Share | | | Fraction of Change | | |
|---|---|---|---|---|---|---|---|
| | | Original | Manipulated | Ratio | More | Tied | Fewer |
| (a) | 0.0098 | 0.0099 | 0.0209 | 211.1% | 0.632 | 0.244 | 0.124 |
| (c) | 0.0098 | 0.0084 | 0.0516 | 614.3% | 0.894 | 0.095 | 0.011 |
| (d) | 0.0098 | 0.0091 | 0.0747 | 820.9% | 0.959 | 0.035 | 0.006 |

Table 11: Results of Outlier Attack under data augmentation.

| Setting | $|Z_1^a|/|Z_1|$ | Compensation Share | | | Fraction of Change | | |
|---|---|---|---|---|---|---|---|
| | | Original | Manipulated | Ratio | More | Tied | Fewer |
| (a) | 0.0098 | 0.0099 | 0.0287 | 289.9% | 0.776 | 0.158 | 0.066 |
| (c) | 0.0098 | 0.0084 | 0.0691 | 822.6% | 0.853 | 0.113 | 0.034 |
| (d) | 0.0098 | 0.0091 | 0.0481 | 528.6% | 0.916 | 0.065 | 0.019 |

## C.5 COMPENSATING ONLY CORRECT PREDICTION

In this section we consider the setup when the `AI Developer` only rewards training data points that are highly influential to validation samples with *correct* predictions. In detail, after training the model and before calculating of compensation share, the `AI Developer` will test the model on a private validation set, select those validation points that are correctly predicted, and only attribute compensation to influential training points for these validation points. We test this on settings (a), (c), and (d). Note that the **Fraction of Change** metric is no longer feasible in this setup since the model trained on the original and manipulated training set may not share the same validation set. The results are shown in Table 12. Overall, our proposed attacks are still highly effective based on the **Ratio** of **Compensation Share**.

Table 12: Results when compensating only correctly predicted data points.

| Attack Type | Setting | $|Z_1^a|/|Z_1|$ | Compensation Share | | |
|---|---|---|---|---|---|
| | | | Original | Manipulated | Ratio |
| Shadow Attack | (a) | 0.0098 | 0.0050 | 0.0382 | 764.0% |
| | (c) | 0.0098 | 0.0090 | 0.0752 | 835.6% |
| | (d) | 0.0098 | 0.0110 | 0.0353 | 320.9% |
| Outlier Attack | (a) | 0.0098 | 0.0050 | 0.0273 | 546.0% |
| | (c) | 0.0098 | 0.0090 | 0.0552 | 613.3% |
| | (d) | 0.0098 | 0.0110 | 0.0227 | 206.4% |

## C.6 ADVERSARIAL ATTACK ON LARGE-SCALE SETUP

In this section, we scale up our experiment and see how the proposed attacks generalize in this large-scale experiment. Specifically, we consider the ResNet-18 model (He et al., 2016) on the Tiny ImageNet dataset (Le & Yang, 2015) with $|Z_0|, |Z_1|) = (50000, 60000)$, and we perturb $|Z_1^a| = 100$ training points. Both the Shadow Attack and Outlier Attack are tested, and the results are shown in Table 13. It is evident that from the **Ratio** of **Compensation Share**, two proposed attacks are still highly effective in a large-scale experiment.

Table 13: Results of Shadow Attack and Outlier Attack under large-scale setup .

| Attack Type | $|Z_1^a|/|Z_1|$ | Compensation Share | | | Fraction of Change | | |
|---|---|---|---|---|---|---|---|
| | | Original | Manipulated | Ratio | More | Tied | Fewer |
| Outlier Attack | 0.0017 | 0.0009 | 0.0056 | 622.2% | 0.456 | 0.498 | 0.046 |
| Shadow Attack | 0.0017 | 0.0009 | 0.0181 | 2011.1% | 0.855 | 0.131 | 0.014 |

# D VISUALIZATION OF THE PERTURBATION

Finally, Figure 3 provides visualizations of two examples of the adversarial perturbed images from MNIST and CIFAR-10. In both cases, the perturbations are barely visible to human eyes, however, they are highly effective in terms of the success of the attack.

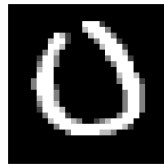
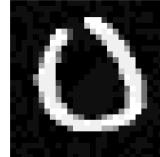
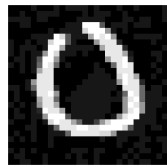

(a) **Original.** Influential for 0 validation data points.

(b) **Shadow Attack.** Influential for 75 validation data points.

(c) **Outlier Attack.** Influential for 105 validation data points.

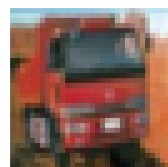
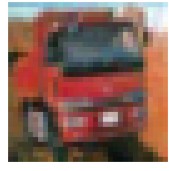
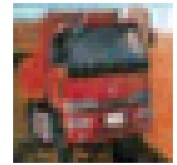

(d) **Original.** Influential for 1 validation data points.

(e) **Shadow Attack.** Influential for 38 validation data points.

(f) **Outlier Attack.** Influential for 29 validation data points.

Figure 3: Visualization of MNIST (*Top*) and CIFAR-10 (*Bottom*), before and after attacks.

