# OpenReview forum: "Adversarial Attacks on Data Attribution"
_ICLR.cc/2025/Conference — ICLR 2025 Poster_

### Official Review · Reviewer_wvst · 2024-10-30

**Soundness:** 2
**Presentation:** 2
**Contribution:** 2
**Rating:** 3
**Confidence:** 4

**Summary:**

Data attribution quantifies the contribution of individual data points during model training. Higher contribution scores indicate that a data point has more value within the training dataset. This paper proposes two adversarial methods—Shadow Attack and Outlier Attack—designed to manipulate data attribution. The authors evaluate these methods against various models and datasets, demonstrating that they can significantly inflate the contribution scores of otherwise benign data points.

**Strengths:**

**Strengths:**

- **Addresses Critical Gap in Research**: A key contribution of this work is that it addresses a crucial research gap. The robustness of data attribution methods has remained largely unexplored. This study is the first to systematically investigate vulnerabilities in data attribution.

- **Significant Attribution Inflation**: The authors introduce two effective attacks that can substantially increase the contribution scores of input samples. The reviewer was impressed by the triple-digit inflation rates achieved, showing the potential impact of these adversarial techniques.

**Weaknesses:**

1. **Threat Model**: The authors describe a black-box threat model, which traditionally implies that the attacker has no information about the model other than query access. However, the authors assume access to the model’s architecture, indicating a grey-box rather than a black-box scenario. It would be beneficial for the authors to revise the text to accurately reflect this nuanced threat model.

2. **Alternative to Creating Outliers**: The paper presents two algorithms designed to create natural-looking outliers with a high impact on the model. Given the complexity of these algorithms, a simpler approach might involve manually mislabeling inputs to create outliers (e.g., labeling a "cat" as a "dog" in a classification task), which would likely produce a similar exaggerated impact on the model. The authors should consider addressing why this more straightforward and cost-effective method was not explored.

3. **Simplistic Pipeline**: The authors assume a straightforward threat model where the defender does not employ methods to detect out-of-distribution (OOD) samples or sanitize inputs. Adversarial samples, such as those generated by the Shadow Attack, are known to be vulnerable to detection and degradation [1]. Testing the attacks against state-of-the-art OOD detection and adversarial input detection methods would help establish their robustness.

4. **Simple Defense**: A basic yet effective defense against these attacks would be data augmentation, such as cropping or shifting images to generate additional samples. This method improves model generalization, a common practice in machine learning, and could reduce the impact of adversarial samples like those introduced by the authors. The authors should demonstrate that including data augmentation would not impact their attack, as it more accurately reflects real-world training environments.

[1] https://arxiv.org/abs/1902.02918

**Questions:**

See above

---

> ### Author Response · Authors · 2024-11-25
>
> > Weakness 1: Black-box assumption of the threat model
>
> We would like to clarify that while we described in total three assumptions in Section 3.2, the two attacks we proposed rely on **different sets** of assumptions.
>
> As explicitly stated in Line 188-189, the Outlier Attack does NOT assume Assumption 2 (Access to data distribution and training algorithm), and it indeed does **NOT** require any information about the model other than query access. So Outlier Attack is indeed a black-box attack method.
>
> It is only the Shadow Attack that requires information related to the training algorithm, including the rough model architecture. However, we have NEVER claimed that Shadow Attack is a black-box attack method in our paper.
>
> To further clarify this point, we have added one sentence at Line 189 to explicitly state that Shadow Attack is a gray-box attack method while Outlier Attack is a black-box attack method.
>
> > Weakness 2: Creating outliers via mislabeling
>
> We kindly refer the reviewer to section 5.2 of our paper. In the “ Data Component To Be Perturbed” paragraph, we have discussed why directly flipping the label of the training data point is not a practical attack strategy.  To recap, the main reasons are:
> 1. Such an outlier with a wrong label often degrades model performance and is likely to be identified as negatively influential;
> 2. It is easy for the AI developer to detect such wrongly-labeled data points.
>
> > Weakness 3: OOD and adversarial defense against proposed attacks.
>
> We appreciate the reviewer’s mention of OOD detection and adversarial defense techniques.
>
> However, we respectfully disagree with the characterization of assuming no adversarial sample detection in the AI developer pipeline as "simplistic." There is an **inherent tradeoff between robustness and utility** in nearly all adversarial defense techniques. Without a clear understanding of potential threats, it is unlikely that AI developers will preemptively adopt adversarial defense strategies in their development pipeline.
>
> To the best of our knowledge, this is **the first systematic study of adversarial attacks on data attribution**. Our study reveals effective yet surprising (especially the Outlier Attack) attack strategies on data attribution. It is precisely the key contribution of this work to alert AI developers that adversarial defense techniques are necessary when integrating data attribution into their pipeline. We also provide insights about how adversaries might exploit their system. This enables developers to **select appropriate defense techniques without excessively comprimising model utility**.
>
> Additionally, we would like to highlight that the “outliers” generated by our attack methods may not resemble the natural outliers typically addressed by conventional OOD detection techniques. Therefore, the effectiveness of these methods in countering our proposed adversarial attacks remains uncertain.
>
> While the development of effective defense techniques is an important avenue for future work, we believe that our study makes a significant contribution as the first systematic investigation of adversarial vulnerabilities in data attribution. By revealing surprisingly simple yet effective attack methods, including one requiring only black-box access to the model, this work lays a critical foundation for further research, even without an extensive exploration of defense strategies.
>
> > Weakness 4: Data augmentation defense against proposed attacks
>
> We appreciate the reviewer’s suggestion of data augmentation as a simple defense against the proposed attacks. We acknowledge that data augmentation is a common step in AI development pipeline and have conducted additional experiments in such setups.
>
> We conducted experiments on the image classification task where we applied cropping and affine-transformation augmentations. We assume that the adversary knows nothing about the augmentation and perturbs their own data as if there is no augmentation when they conduct shadow training in Shadow Attack or generate adversarial examples in Outlier Attack. The results are shown below and both attack methods are still highly effective under data augmentation.

---

> ### Author Response · Authors · 2024-11-25
>
> Results for Shadow Attack:
> | Setting | $Z_1^{a}/Z_1$ | Compensation Share Original | Compensation Share Manipulated | Ratio (%)  | Fraction of Change More | Fraction of Change Tied | Fraction of Change Fewer |
> |---------|-------------------|-----------------------------|--------------------------------|------------|--------------------------|--------------------------|--------------------------|
> | (a)     | $0.0098$          | $0.0099$                   | $0.0209$                       | $211.1\%$  | $0.632$                 | $0.244$                 | $0.124$                 |
> | (c)     | $0.0098$          | $0.0084$                   | $0.0516$                       | $614.3\%$  | $0.894$                 | $0.095$                 | $0.011$                 |
> | (d)     | $0.0098$          | $0.0091$                   | $0.0747$                       | $820.9\%$  | $0.959$                 | $0.035$                 | $0.006$                 |
>
>
>
> Results for Outlier Attack:
> | Setting | $Z_1^{a}/Z_1$ | Compensation Share Original | Compensation Share Manipulated | Ratio (%)  | Fraction of Change More | Fraction of Change Tied | Fraction of Change Fewer |
> |---------|-------------------|-----------------------------|--------------------------------|------------|--------------------------|--------------------------|--------------------------|
> | (a)     | $0.0098$          | $0.0099$                   | $0.0287$                       | $289.9\%$  | $0.776$                 | $0.158$                 | $0.066$                 |
> | (c)     | $0.0098$          | $0.0084$                   | $0.0691$                       | $822.6\%$  | $0.853$                 | $0.113$                 | $0.034$                 |
> | (d)     | $0.0098$          | $0.0091$                   | $0.0481$                       | $528.6\%$  | $0.916$                 | $0.065$                 | $0.019$                 |

---

> > ### Comment · Reviewer_wvst · 2024-12-01
> >
> > I appreciate the authors taking the time to read my review and conducting additional experiments. However, my primary concern remains: the work does not offer any new insights. It simply reaffirms that models are vulnerable to adversarial attacks—a well-established finding with extensive prior research. My review aimed to suggest potential directions for strengthening the work. While the authors have addressed Weakness #4, Weaknesses #2 and #3 remain unaddressed. To enhance the contribution of this work, I would expect either new insights from the proposed method or a more comprehensive evaluation across various defenses. Consequently, I will maintain my original score.

---

> > > ### Author Response · Authors · 2024-12-01
> > >
> > > Thank you for the follow up.
> > >
> > >
> > > > the work does not offer any new insights. It simply reaffirms that models are vulnerable to adversarial attacks.
> > >
> > > We would like to respectfully clarify a key distinction: **adversarial robustness of the ML models** and the **adversarial robustness of data attribution applied to ML models** are fundamentally different topics. Whlile the former has been widely studied, our paper focuses on the latter, addressing a crucial research gap. This point has been acknowledged in the Reviewer **wvst**'s own review:
> > >
> > > *"A key contribution of this work is that it addresses a crucial research gap. The robustness of data attribution methods has remained largely unexplored. This study is the first to systematically investigate vulnerabilities in data attribution."*
> > >
> > > Other reviewers have also acknowledged that our "threat model is well defined" (Reviewer **4aYY**) and the "proposed attacks are novel" (Reviewer **pQy1**).
> > >
> > > Therefore, we respectfully disagree that our work *"simply reaffirms that models are vulnerable to adversarial attacks"*.
> > >
> > > > Weakness #2
> > >
> > > In Weakness #2, the reviewer suggested a simple alternative attack method by *"manually mislabeling inputs to create outliers (e.g., labeling a "cat" as a "dog" in a classification task)"*. In both **our original submission (Section 5.2)** and our response, we have clarified why this simple strategy is NOT practical:
> > >
> > > 1. Such an outlier with a wrong label often degrades model performance and is likely to be identified as negatively influential instead of positviely influential that contributes to the compensation share;
> > > 2. It is easy for the AI developer to detect such wrongly-labeled data points.
> > >
> > > > Weakness #3
> > >
> > > In Weakness #3, the reviewer suggested OOD detection or adversarial defense methods are implemented in typical machine learning pipelines. In our response, we have clarified that
> > >
> > > 1. There is an **inherent tradeoff between robustness and utility** in nearly all adversarial defense techniques. Without a clear understanding of potential threats, it is unlikely that AI developers will preemptively adopt adversarial defense strategies in their development pipeline.
> > > 2. Given that this is **the first systematic study of adversarial attacks on data attribution**, it is precisely the key contribution of this work to alert AI developers that adversarial defense techniques are necessary when integrating data attribution into their pipeline.
> > > 3. We also provide insights about how adversaries might exploit their system. This enables developers to **select appropriate defense techniques without excessively comprimising model utility**.
> > >
> > > > To enhance the contribution of this work, I would expect either new insights from the proposed method or a more comprehensive evaluation across various defenses.
> > >
> > > We would like to highlight that the insights provided by our proposed attacks are novel and significant in the context of adversarial vulnerabilities in data attribution. For the Outlier Attack, we further offered **a novel theoretical analysis** of why data attribution mechanisms are susceptible to adversarial examples.
> > >
> > > Furthermore, it is common for studies introducing new adversarial vulnerabilities to focus primarily on the attack rather than exhaustively evaluating all possible defenses. While the development of effective defense techniques is an important avenue for future work, we believe that our study makes a significant contribution as the first systematic investigation of adversarial vulnerabilities in data attribution.

---

### Official Review · Reviewer_4aYY · 2024-11-01

**Soundness:** 2
**Presentation:** 3
**Contribution:** 3
**Rating:** 6
**Confidence:** 3

**Summary:**

This paper introduces two data attribution attack methods, Shadow Attack and Outlier Attack, that aim to inject new training data that maximizes the influence of target model outputs. Both attacks optimize over a compensation share function, which quantifies the influence of a subset of the training data to the target models predictions on test data. Shadow Attack assumes the adversary has access to data distributions and the training algorithm, and uses gradient ascent on training data to generate data points. Outlier Attack assumes black-box query access to the target model, and uses black-box image attack algorithms on training data to generate data points. The attacks are evaluated on image classification and text generation tasks using metrics that measure the change in compensation share before and after adversarial data is introduced.

**Strengths:**

Thank you for submitting to ICLR 2025. I enjoyed reading about the compensation function used for the adversarial objective in the threat model section. I have listed several points that I find the paper do well below:
- The threat model is well defined for both attack methods. Assumptions made of target model and training/testing data access is clearly stated. The periodic data contribution attack setting between AI Developers and Data Providers is realistic.
- The adversarial objective is very clear and the compensation function defined in the paper captures it in a novel way. Additionally, the Shadow Attack and Outlier Attack methodology is designed within the limits of the threat model.
- Baseline evaluation helps articulate the performance of the attack methods. Since the compensation share metric is new, the baseline evaluation puts into context how performant the attack is.

**Weaknesses:**

The problem discussed is very interesting and deserves more attention from researchers and developers from both academia and industry. It is worth noting the effort made by the authors in designing novel attack methodologies. However, I believe there is still room for improvement, especially in making claims (e.g., attack effectiveness and efficiency) more concrete and clear and including more baseline evaluations that shed light on the performance of proposed attack under the given threat model. More details can be found below.
- The paper lacks well-defined research questions in the evaluation. It can be inferred that sections on Shadow Attack and Outlier Attack measure effectiveness, but it would improve the paper if there were claims made that the result would prove/disprove.
- Another baseline metric would be great to understand the performance of the attack under the defined threat model. While the random perturbation baseline helps contextualize the compensation share ratio of the attacks, another baseline assuming a white-box threat model (i.e., the adversary has access to the target model and its parameters) would highlight the performance under the defined threat model with more limitations.

**Questions:**

Thanks to the authors for the submission to ICLR 2025. I enjoyed reading about the compensation function used for the adversarial objective in the threat model section. Most of my suggestions are already mentioned in the above Weakness section. Some questions (might be open-ended) are listed below. It would be great if more discussion or clarification on the following questions could be added to the paper:
- How is distortion budget measured for both image classification and text generation tasks? What norm is used for images and is the budget consistent with other white-box and black-box attack baselines?
- Analyzing how data attribution methods affect attack performance would strengthen the study of the attacks. For example, if one were to experiment with the Influence Function, Data Shapley, and TRAK methods for a particular {task, dataset, target model} configuration, how do the attacks perform?
- If we increase or decrease the size of Z_0 and Z_1 for each setting, would this make the attack more successful (i.e., how does performance change with respect to the amount of data)?

---

> ### Author Response · Authors · 2024-11-25
>
> We thank you for your detailed insights and questions.
>
> > Weakness 1: The paper lacks well-defined research questions in evaluation.
>
> Our experiment results primarily and effectively support the main argument: the data attribution score can be artificially increased by careful design of adversarial attack methods which we show in the paper. We have added a description at the beginning of Section 6 to clarify it.
>
> > Weakness 2: White-box attack can be added as another baseline.
>
> We implemented the white-box attack where the adversary has full knowledge of the model on the image classification task. The results show that the effectiveness of the black-box attack is slightly worse but comparable to the white-box attack, further highlighting the effectiveness of the proposed methods.
>
> | Setting | Attack Method | Compensation Share Original | Compensation Share Manipulated | Ratio (%)  | Fraction of Change More | Fraction of Change Tied | Fraction of Change Fewer |
> |---------|---------------|--------------------|--------------------------|------------|---------------|---------------------|--------------------------|
> | (a)     | FGSM          | $0.0098$                   | $0.0649$                       | $662.2\%$  | $0.990$                 | $0.007$                 | $0.003$                 |
> | (d)     | PGD           | $0.0095$                   | $0.0222$                       | $233.7\%$  | $0.689$                 | $0.204$                 | $0.107$                 |
>
>
>
> > Question 1: The distortion/perturbation budget of our attack methods.
>
> We have closely followed the existing literature when setting the perturbation budget. Please check the relevant literature: ZOO [1], Simba [2], TextFooler [3]. We also had a detailed description in Appendix B.3 in our original submission.
>
> [1] ZOO: Zeroth Order Optimization Based Black-box Attacks to Deep Neural Networks without Training Substitute Models. https://dl.acm.org/doi/abs/10.1145/3128572.3140448
>
> [2] Simple Black-box Adversarial Attacks. https://arxiv.org/pdf/1905.07121
>
> [3] Is BERT Really Robust? A Strong Baseline for Natural Language Attack on Text Classification and Entailment. https://arxiv.org/pdf/1907.11932
>
>
> > Question 2: Experiment with different data attribution methods on the same setup.
>
> We thank the reviewer for the suggestion. We have extended experiments with Influence Function and TRAK, in addition to the original Data Shapley, on experimental setup (b). Note that Data Shapley is much more computationally intensive than Influence Function and TRAK, which is not feasible on large datasets, especially under constrained computational resources. The results below show that the proposed attack methods are highly effective for all three data attribution methods on the same setup. The difference brought by different data attribution methods is relatively small in comparison to the absolute increase of compensation share.
>
> The Results for Shadow Attack:
>
> | Attribution Method   | $Z_1^{a}/Z_1$ | Compensation Share Original | Compensation Share Manipulated | Ratio (%)  | Fraction of Change More | Fraction of Change Tied | Fraction of Change Fewer |
> |-----------------------|-------------------|-----------------------------|--------------------------------|------------|--------------------------|--------------------------|--------------------------|
> | Data Shapley          | $0.0352$         | $0.0152$                    | $0.0435$                       | $286.2\%$  | $0.533$                 | $0.333$                 | $0.134$                 |
> | Influence Function    | $0.0352$         | $0.0496$                    | $0.1004$                       | $202.4\%$  | $0.980$                 | $0.000$                 | $0.020$                 |
> | TRAK                  | $0.0352$         | $0.0392$                    | $0.0936$                       | $238.8\%$  | $0.820$                 | $0.100$                 | $0.080$                 |
>
>
>
>
>
> The Results for Outlier Attack:
> | Attribution Method   | $Z_1^{a}/ Z_1$ | Compensation Share Original | Compensation Share Manipulated | Ratio (%)  | Fraction of Change More | Fraction of Change Tied | Fraction of Change Fewer |
> |-----------------------|-------------------|-----------------------------|--------------------------------|------------|--------------------------|--------------------------|--------------------------|
> | Data Shapley          | $0.0250$         | $0.0112$                    | $0.0218$                       | $194.6\%$  | $0.440$                 | $0.380$                 | $0.180$                 |
> | Influence Function    | $0.0250$         | $0.0186$                    | $0.0442$                       | $237.6\%$  | $0.920$                 | $0.060$                 | $0.020$                 |
> | TRAK                  | $0.0250$         | $0.0160$                    | $0.0412$                       | $257.5\%$  | $0.780$                 | $0.180$                 | $0.040$                 |

---

> ### Author Response · Authors · 2024-11-25
>
> > Question 3: Attack performance under different sizes of $Z_0$ and $Z_1$.
>
> We have conducted experiments for both attack methods with increased and decreased sizes of $Z_0$ and $Z_1$. The results can be found in Appendix C.1, Table 5 and 6. For convenience, we summarize the results here as well. We see that the results demonstrate the effectiveness of our methods across different sizes of the dataset. Overall, with a few exceptions, the **Ratio** of the **Compensation Share** appears to become even higher when the dataset size is larger.
>
> The results of Shadow Attacks:
>
> | Setting | $(Z_0, Z_1)$     | Compensation Share Original | Compensation Share Manipulated | Ratio (%)  | Fraction of Change More | Fraction of Change Tied | Fraction of Change Fewer |
> | ------- | ---------------- | ------------------- | ----------------------- | ---------- | -------------- | -------------- | -------------------- |
> | (a)     | $(5000, 6000)$   | $0.0153$         | $0.0764$           | $499.3\%$  | $0.996$                 | $0.003$                 | $0.001$                  |
> | (a)     | $(10000, 11000)$ | $0.0098$        | $0.0477$       | $456.1\%$  | $0.955$                 | $0.038$                 | $0.007$                  |
> | (a)     | $(15000, 16000)$ | $0.0050$     | $0.0373$           | $746.0\%$  | $963$                   | $0.036$                 | $0.001$                  |
> | (c)     | $(5000, 6000)$   | $0.0168$   | $0.0539$              | $320.7\%$  | $0.857$                 | $0.129$                 | $0.014$                  |
> | (c)     | $(10000, 11000)$ | $0.0112$             | $0.0467$            | $417.0\%$  | $0.781$                 | $0.195$                 | $0.024$
> | (c)     | $(15000, 16000)$ | $0.0002$               | $0.0062$      | $3100.0\%$ | $0.431$                 | $0.568$                 | $0.001$                  |
> | (d)     | $(5000, 6000)$   | $0.0206$                | $0.0413$       | $200.7\%$  | $0.696$                 | $0.174$                 | $0.130$                  |
> | (d)     | $(10000, 11000)$ | $0.0095$      | $0.0213$    | $217.3\%$  | $0.655$                 | $0.259$                 | $0.086$                  |
> | (d)     | $(15000, 16000)$ | $0.0057$   | $0.0092$      | $161.5\%$  | $0.264$                 | $0.616$                 | $0.120$                  |
>
> The results of Outlier Attacks:
> | Setting | $(Z_0, Z_1)$     | Compensation Share Original | Compensation Share Manipulated | Ratio (%)  | Fraction of Change More | Fraction of Change Tied | Fraction of Change Fewer |
> | ------- | ---------------- | --------------------------- | ------------------------------ | ---------- | ----------------------- | ----------------------- | ------------------------ |
> | (a)     | $(5000, 6000)$   | $0.0153$                    | $0.0747$                       | $488.9\%$  | $0.998$                 | $0.002$                 | $0.000$                  |
> | (a)     | $(10000, 11000)$ | $0.0098$                    | $0.0631$      | $643.9\%$  | $0.980$                 | $0.017$                 | $0.003$                  |
> | (a)     | $(15000, 16000)$ | $0.0050$                    | $0.0400$    | $800.0\%$  | $0.964$                 | $0.036$                 | $0.000$                  |
> | (c)     | $(5000, 6000)$   | $0.0168$                    | $0.0334$     | $198.8\%$  | $0.720$                 | $0.241$                 | $0.039$                  |
> | (c)     | $(10000, 11000)$ | $0.0112$                    | $0.0668$ | $596.4\%$  | $0.799$                 | $0.173$                 | $0.028$                  |
> | (c)     | $(15000, 16000)$ | $0.0002$   | $0.0051$                       | $2550.0\%$ | $0.397$                 | $0.603$                 | $0.000$                  |
> | (d)     | $(5000, 6000)$   | $0.0206$    | $0.0411$      | $199.5\%$  | $0.761$                 | $0.133$                 | $0.106$                  |
> | (d)     | $(10000, 11000)$ | $0.0095$    | $0.0176$   | $185.2\%$  | $0.452$                 | $0.354$                 | $0.084$                  |
> | (d)     | $(15000, 16000)$ | $0.0057$     | $0.0219$     | $384.2\%$  | $0.731$                 | $0.192$                 | $0.077$                  |
> | (e)     | $(5000, 6000)$   | $0.0013$       | $0.0035$       | $262.9\%$  | $0.392$                 | $0.461$                 | $0.147$                  |
> | (e)     | $(10000, 11000)$ | $0.0016$      | $0.0064$        | $400.0\%$  | $0.420$  | $0.507$                 | $0.073$                  |
> | (e)     | $(15000, 16000)$ | $0.0029$     | $0.0214$    | $737.9\%$  | $0.400$                 | $0.543$                 | $0.057$                  |
>
> Minor Edit: the 11 row 3 column of the result of Outlier Attacks in the initial post has a typo: the number $0.0031$ should be $0.0013$ which has been updated correctly now. Note that the Ratio value (11 row 5 column) is originally calculated correctly based on the correct value and remains accurate.

---

> > ### Comment · Reviewer_4aYY · 2024-12-01
> > **Response to Authors**
> >
> > I appreciate the authors' thorough response to the concerns and the revisions made to the paper. I believe the paper has improved in certain aspects. Below, I address the revisions made in response to the two weaknesses I previously identified.
> > - **Weakness 1**: Lack of Well-Defined Research Questions
> >
> > In my initial feedback, I highlighted the need for clear, well-defined research questions to provide structure for your study. While I appreciate the authors' attempt to address this by including a sentence at the beginning of Section 6, I find this addition is insufficient to clarify the broader scope and contributions of the paper.
> > - **Weakness 2**: Lack of Baselines for Evaluation
> >
> > I appreciate the authors' efforts to address concerns regarding the lack of baselines. The inclusion of the white-box setting has strengthened the argument and provided a better understanding of how the proposed attacks perform with more limited access to the victim model. Thank you for making this improvement- it is a valuable addition to the paper.
> >
> > In summary, while the revisions addressing baselines are well executed, the addition of concrete research questions requires further development. I hope the authors will consider expanding on this aspect for the next version. Thank you again for the efforts in revising the paper.

---

> > > ### Author Response · Authors · 2024-12-01
> > >
> > > Dear Reviewer 4aYY,
> > >
> > > Thank you for the follow-up reply and for acknowledging that we have addressed Weakness 2.
> > >
> > > Regarding Weakness 1, the original comment was *"The paper lacks well-defined research questions in the **evaluation**"* We interpreted this as a comment on the clarity of the goal of the experiments. While we agree that explicitly clarifying these goals would improve the paper, the objective of our experiments is straightforward: to measure the increase in compensation share achieved through the proposed attacks. To address this, we included an additional sentence to make the goal explicit.
> > >
> > > We apologize if we misinterpreted this comment. Motivated by your follow-up comment on the contributions and scope of the paper, we plan to add the following paragraph into our Introduction (at line 049 after the third paragraph) to explicitly articulate the research questions:
> > >
> > > **In this paper, we aim to address the following question: Is it possible for a malicious data contributor to unfairly inflate their compensation share by adversarially manipulating the data they provide? If so, what strategies enable such manipulation, and what level of knowledge about the AI system must they possess to succeed?**
> > >
> > > Our paper has successfully answerd all these questions. The answer to the first question is affirmative. And we have provided two effective strategies supported by our comprehensive empirical evaluation. We have also provided formal assumptions about the knowledge these two strategies required, outlined in our threat model.
> > >
> > > We hope this addition addresses your concern about clarifying the contributions and scope of our paper.
> > >
> > > Finally, we would like to kindly ask the reviewer if there are any remaining major concerns or drawbacks in our paper that might prevent it from being acceptable. We would greatly appreciate it if the reviewer could point them out, allowing us the opportunity to address them.
> > >
> > > Thank you again for yout time and effort in reviewing our submission!

---

> > > > ### Comment · Reviewer_4aYY · 2024-12-02
> > > > **Response to Authors (2/2)**
> > > >
> > > > I appreciate the authors' timely and clear response to the outstanding concern. The additional goal statement made in the intro makes the research questions clearer in the evaluation.
> > > >
> > > > The new sentence added to the introduction shows that the research questions coincide with (1) the effectiveness of the attack measured by compensation share differences and (2) how these methods compare to different levels of access to the victim model (varying threat models). Despite this not thoroughly states in section 6, this new statement will suffice to bump up the score of the paper.

---

### Official Review · Reviewer_T7Fd · 2024-11-04

**Soundness:** 3
**Presentation:** 2
**Contribution:** 2
**Rating:** 6
**Confidence:** 4

**Summary:**

This paper studies adversarial attacks to data attribution based compensation mechanism.

Two empirical attacks are proposed.
1. Shadow attack, which assumes access to data distribution & training algorithm, creates a surrogate data attribution system and adversarially optimize the compensations with the surrogate.
2. Outlier attack, which assumes access to the black-box query of the model and directly optimizes to minimize the confidence of the model to generate outlier data, which could correspond to higher compensation.

**Strengths:**

The paper is overall clear and easy to follow.

Since the adversarial robustness of data attribution is a reasonable subject, the significance is not considered an issue.

**Weaknesses:**

The experimental objective function of the adversarial attack to data attribution requires much better justifications.

The paper motivates the study of the robustness of data attribution with data attribution based compensation.
However, it is natural that such compensation designs reward only for data contributing to good/desired behaviors: For example, they might allocate compensations only based on attributions for **correct** predictions (these could be from a prepared validation/test set or from user feedbacks). Alternatively, they might withdraw compensations or even incur penalties for **incorrect/undesired** predictions.

A concern is that in such, arguably more reasonable settings, neither of the proposed attacks will be in fact effective. For shadow attack, only data attributed greatly for correct predictions will be awarded, which is totally fine as the "attack" is in some sense actually making the data more valuable, with efforts. For outlier attacks, the outliers are less likely to be attributed if the prediction is correct and consistent with regular training data.

**Questions:**

See Weaknesses for my primary concerns, which is in fact somewhat critical to make decisions regarding the impact of the observations.

---

> ### Author Response · Authors · 2024-11-25
>
> We thank the reviewer for the insightful comments and questions.
>
> We would like to first clarify a potential misinterpretation of our compensation mechanism. While the compensation mechanism indeed involved the data points with incorrect/undesired predictions, we highlight that the attribution score is calculated with respect to the loss on these data points, i.e., the top influential data points are always the ones contributing to the **loss reduction** (regardless the correctness of the predictions). Even when the model makes incorrect predictions on certain validation data points, the training data points that contribute to **dragging the model towards the correct prediction direction** could still be considered **valuable**.
>
> To fully address the reviewer’s concern, we also conducted an experiment in which the AI developer would only reward the correct predictions on a private validation set on the image classification task. The result is presented as follows, and the increase in compensation share exceeds 200%, which shows the effectiveness of our method in a compensation mechanism where only correct predictions are attributed.
>
> Results for Both Attacks:
> | Attack Type     | Setting | $Z_1^{a}/Z_1$ | Compensation Share Original | Compensation Share Manipulated | Ratio (%)  |
> |------------------|---------|-------------------|-----------------------------|--------------------------------|------------|
> | Shadow Attack    | (a)     | $0.0098$          | $0.0050$                    | $0.0382$                       | $764.0\%$  |
> |         Shadow Attack           | (c)     | $0.0098$          | $0.0090$                    | $0.0752$                       | $835.6\%$  |
> |        Shadow Attack            | (d)     | $0.0098$          | $0.0110$                    | $0.0353$                       | $320.9\%$  |
> | Outlier Attack   | (a)     | $0.0098$          | $0.0050$                    | $0.0273$                       | $546.0\%$  |
> |     Outlier Attack                | (c)     | $0.0098$          | $0.0090$                    | $0.0552$                       | $613.3\%$  |
> |      Outlier Attack               | (d)     | $0.0098$          | $0.0110$                    | $0.0227$                       | $206.4\%$  |
>
>
>
>
> The consistency between the two compensation mechanisms is not surprising, as both mechanisms aim to reward training data points that contribute to dragging the model towards the correct prediction direction.
>
> Finally, we would like to highlight that in the Outlier Attack, the outliers are **correctly labeled** samples that do not look similar to the rest of the training dataset. Intuitively, this is why these outliers are more likely to be attributed as influential data points helping correct predictions.

---

> > ### Comment · Reviewer_T7Fd · 2024-11-25
> >
> > Thank you for that clarification.
> >
> > Correspondingly, as I also raised in my original review, the issue being that: Only data attributed greatly for correct predictions will be awarded, which is totally fine as the "attack" is in some sense actually making the data more valuable, with efforts.
> >
> > For this to be an attack targeting data attribution rather than a totally fine extra work to improve data utility, I would like to ask the authors the following:
> > **Does the attack not increase the target model performance/decrease loss (on the corresponding distribution or validation) when it is getting a bigger share of compensation?**
> >
> > Could you, for example, report the performance of the target models, before/after such attacks on the same (validation) set in addition to the already reported increase of compensations?
> > I suppose these numbers should be already available somewhere without re-doing experiments.

---

> ### Author Response · Authors · 2024-11-25
> **Follow-up response to Reviewer T7Fd**
>
> > Follow-Up Question:  Does the attack not increase the target model performance/decrease loss (on the corresponding distribution or validation) when it is getting a bigger share of compensation?
>
> We thank the reviewer for this insightful question. The attack does NOT increase the target model performance. Please see below a summary of the image classification accuracy across different settings.
>
> | Setup/Accuracy       | Original (w/o Manipulation) | Shadow Attack | Outlier Attack |
> |-----------------------|-----------------------|---------------|----------------|
> | (a)                  | 89.4%                | 89.2%         | 89.4%          |
> | (c)                  | 98.1%                | 97.9%         | 97.8%          |
> | (d)                  | 72.2%                | 70.6%         | 71.7%          |
>
> In comparison to the model trained on the original clean data without manipulation, the model trained on the data manipulated by the attacks has similar or slightly worse performance in all settings.
>
> Intuitively, the Compensation Share is a **“zero-sum”** metric among all the training data points. The attacks increase the Compensation Share of the training data points contributed by the adversary, which, by definition, means **the Compensation Share of the rest of the training data points is decreased**. The increase in Compensation Share of a subset of the training data does not necessarily lead to the overall performance gain.
>
> We hope the additional results and the clarification on the intuition can successfully address your concern. We will also update the draft later to include this information.

---

> > ### Comment · Reviewer_T7Fd · 2024-11-25
> >
> > Yes, including the information will be important and please make sure to do so later (and include results for other settings as well).
> >
> > I am somewhat ok with giving a positive score for this paper now after seeing the additional numbers, so I will temporarily raise my score to 6. I will later further incorporate the opinions of other reviewers.
> >
> > That being said, for setting (d), about 70% accuracy with resnet-18 on CIFAR-10 seems low even with 10k samples; I strongly encourage authors to double-check the training configs for the corresponding settings, making sure this is not due to implementation errors/improper hyper-params (which may weaken the implications of the experiments).

---

> ### Author Response · Authors · 2024-11-30
> **Follow-up response to Reviewer T7Fd**
>
> > Follow-Up Question on Model Performance
>
> Thank you for raising this concern. Our experiments did not apply regularization techniques such as data augmentation, which led to a performance gap compared to the optimal one reported in the literature. The impact of such regularization techniques may be particularly important given the smaller training data size we used. To address your concern, we conducted an additional experiment with ResNet-18 on CIFAR-10 with $Z_0 = 147000$, $Z_1 = 150000$ (the full training data), with data augmentation. We only have results for Outlier Attack at this point, as Shadow Attack experiments take much more time. The results are summarized below.
> | Original Accuracy | Accuracy After Outlier Attack | Original Compensation Share | Compensation Share After Outlier Attack | Ratio (%)  |
> |--------------------|-------------------------------|-----------------------------|-----------------------------------------|------------|
> | $ 93.8\%$              | $ 93.4\%$                            | $0.00182$                      |$ 0.02385 $                                 | $1310.4 $    |
>
> We first observed that the accuracy of the model trained on the original data without manipulation is 93.8\%, which is consistent with the literature. The model trained on the data after manipulation by the attack has slightly lower accuracy, which is consistent with our previous observations on smaller data settings. Meanwhile, the ratio of compensation share change shows that our method is even more effective in this standard ResNet training setup on CIFAR-10. We hope this can address your concern, and we will also update the draft to include this result.

---

> ### Comment · Reviewer_T7Fd · 2024-11-30
>
> Here is my confusion:
> I believe CIFAR-10 has only 60k samples in total (50k of which are training samples). Could the authors elaborate how you define a setting with 150k samples (i.e. z_0=147k & z_1=150k)?
>
> Also providing details such as what augmentations are used would help.

---

> ### Author Response · Authors · 2024-11-30
> **Follow-up response to Reviewer T7Fd**
>
> We apologize for the confusion. We should have written $Z_0 = 49000$ and $Z_1=50000$. The 147k and 150k are the sizes after augmentation. We applied two data augmentations per sample (flipping + rotation and flipping + cropping), thus the total size was tripled. Below is the exact PyTorch function for our data augmentation:
>
> The first augmentation: transforms.RandomVerticalFlip(), transforms.RandomRotation(15)
>
> The second augmentation: transforms.RandomHorizontalFlip(), transforms.RandomCrop(32, padding=4)

---

### Official Review · Reviewer_pQy1 · 2024-11-04

**Soundness:** 3
**Presentation:** 3
**Contribution:** 3
**Rating:** 6
**Confidence:** 3

**Summary:**

This paper introduces a novel adversarial attack on data attributions, where the adversary aims to maximize the compensation share received by the adversary. The authors set a few restrictions and assumptions on the adversary's capabilities and the target model. Two different attacks are proposed based on different choices of assumptions. The shadow attack is based on the assumption that the adversary has access to the distribution of the training data and the validation data, while the outlier attack is based on the assumption that the adversarial can query the model to get the data attribution values. The experimental results demonstrate that both the shadow attack and the outlier attack are highly effective in manipulating data attribution methods, leading to substantial increases in the attacker's compensation share.

**Strengths:**

1. The paper is well-organized and easy to follow. The authors clearly define the adversary's objective, capabilities, and assumptions, which helps readers understand the problem setting.
2. The proposed attacks are novel. The shadow attack and the outlier attack are based on certain scenarios and are effective in manipulating data attribution methods.
3. The experimental results are comprehensive. The authors evaluate the proposed attacks on multiple datasets and data attribution methods, demonstrating the effectiveness of the attacks in practice.

**Weaknesses:**

1. The surrogate objective used to replace the discrete compensation share is not well justified. The authors should provide more insights into why the surrogate objective is a good approximation of the compensation share. Is the surrogate itself already a reasonable to represent the compensation share?
2. The paper emphasizes the validity of the assumptions using the examples of large language models, while the experiments are mostly conducted on relatively smaller size datasets (*e.g.*, MNIST and CIFAR-10). It would be more convincing if the authors could provide more scalability analysis on larger datasets.

**Questions:**

See weaknesses.

---

> ### Author Response · Authors · 2024-11-25
>
> We thank the reviewer for the acknowledgment of our work and the insightful comments.
>
> > Weakness 1: Justifying the surrogate objective.
>
> The original compensation share objective measures the ratio that a specific subset of training data (contributed by the adversary) present in the top-k most influential set, while the surrogate objective measures the sum of the (proxy) influence scores of this training subset. Intuitively, these two objectives are aligned, while the surrogate objective is easier to optimize.
>
> Furthermore, we highlight that in all of our experiments, **the evaluation metric is the original compensation share**, the fact that optimizing our surrogate objective leads to a good performance in the original compensation share metric suggests that our surrogate objective approximates the original compensation share reasonably well.
>
>
> > Weakness 2: Larger-scale experiments needed due to emphasis on LLM setup in Assumption 1
>
> First, we would like to clarify that Assumption 1 only assumes that the training data collected from a previous time step will be largely reused in the next time step. While we mentioned LLMs and recommender systems as examples, this assumption is not specific to LLMs. This assumption may even better hold for smaller-scale applications where the training data are expensive to get, as in such cases, it becomes more likely for the data to be reused in the next time step.
>
> Nevertheless, we agree that our paper could be further strengthened by larger-scale experiments. We have conducted a new experiment on the Tiny Imagenet dataset (see the table below), and we observe that both attack methods are still very effective.
>
> | Attack Type     | $Z_1^a / Z_1$ | Compensation Share Original | Compensation Share Manipulated | Ratio (%)  | Fraction of Change More | Fraction of Change Tied | Fraction of Change Fewer |
> |------------------|------------------|-----------------------------|--------------------------------|------------|--------------------------|--------------------------|--------------------------|
> | Outlier Attack   | $0.0017$        | $0.0009$                    | $0.0056$                       | $622.2%$  | $0.456$                 | $0.498$                 | $0.046$                 |
> | Shadow Attack    | $0.0017$        | $0.0009$                    | $0.0181$                       | $2011.1%$ | $0.855$                 | $0.131$                 | $0.014$                 |

---

### Author Response · Authors · 2024-11-25
**Message To All Reviewers**

We thank all the reviewers for their insightful comments and suggestions. We apologize for the delayed response as we have conducted a series of additional experiments to fully address the reviewer comments. In particular, we have 1) conducted experiments on a larger dataset, Tiny ImageNet; 2) evaluated the proposed attacks when data augmentation is used in training the models; 3) included a white-box reference baseline; 4) evaluated the proposed attacks with a slightly different compensation mechanism suggested by Reviewer T7Fd; 5) and other sensitivity analyses experiments. In all cases, we have once again confirmed that the proposed attacks are highly effective.

We have addressed all the comments in the detailed individual response to each reviewer, as well as updated our paper draft to reflect the changes. Here we would like to highlight a couple of key points in our response.

First, to the best of our knowledge, this is **the first systematic study of adversarial attacks on data attribution**. By revealing surprisingly simple yet effective attack methods, including one requiring only black-box access to the model, this work lays a critical foundation for further research. While Reviewer **wvst** treats the lack of defense techniques in the threat model as a major drawback of this work, Reviewers **4aYY** and **pQy1** have acknowledged the soundness of our threat model. We believe that without a clear understanding of potential threats, the AI developers will usually not preemptively adopt defense techniques due to the robustness and utility trade-off. Therefore, we believe this work presents sufficient contributions even without the development of defense techniques.

Second, we would like to highlight that the proposed compensation mechanism is indeed designed to reward training data points that contribute to correct/desired model predictions (as opposed to rewarding both correct and incorrect predictions). As this is the main (and almost only) concern of Reviewer **T7Fd**, we hope our clarification and additional experiments can sufficiently address the reviewer’s concern.

We thank the reviewers again for their efforts in evaluating our work. And we are more than happy to provide further information if there is any further concern.

---

### Meta-Review · Area_Chair_mQUn · 2024-12-25

**Metareview:**

This paper proposes an adversarial attack on the data-attribution-based compensation mechanism.
The paper introduces two attacks, Shadow Attack is a gray-box attack method while Outlier Attack is a black-box attack method.
The reviewers acknowledge that the paper is well-written and the threat model is well-defined for both attack methods.
Overall, the reviewers think this is an important and interesting problem, and the proposed method is novel. The empirical results are reasonably good.  Given the novelty, I recommend accept.

**Additional Comments On Reviewer Discussion:**

The authors did a good job addressing most of the reviewers' concerns. One exception is reviewer wvst, who insisted the paper does not offer any new insights because it simply reaffirms that models are vulnerable to adversarial attacks. I disagree with this reviewer because this paper studies a novel problem. The reviewer's own review also acknowledged that "A key contribution of this work is that it addresses a crucial research gap. The robustness of data attribution methods has remained largely unexplored." Therefore, I tone down the importance of the comments from this reviewer.

---

### Decision · Program_Chairs · 2025-01-22

Accept (Poster)